# Conformal Risk-Averse Decision Making with Action Conditional Guarantee

Zihan Zhu [1] [*]   Shayan Kiyani [1] [*]   George Pappas [1]   Hamed Hassani [1]

## Abstract

Reliable decision-making pipelines powered by machine learning models require uncertainty quantification (UQ) methods that come with explicit safety guarantees. Conformal prediction provides such UQ by wrapping ML predictions into prediction sets, and recent work by Kiyani et al. (2025b) established that these sets can be translated into optimal risk-averse decision policies—yet only inheriting marginal safety guarantees. We generalize and strengthen their results by (i) introducing action-conditional conformal prediction, which yields safety guarantees conditioned explicitly on each action taken by the decision maker, (ii) showing that action-conditional prediction sets serve as a proxy for the feasible decision space for risk-averse decision makers aiming to optimize action-conditional value-at-risk, and (iii) proposing a principled finite-sample algorithm based on pinball-loss minimization, connecting the framework of Gibbs et al. (2025) to action-conditional guarantees. Experiments on two real-world datasets confirm that our approach significantly improves action-conditional performance over several conformal baselines.

## 1. Introduction

In many high-stakes scenarios—such as clinical medicine, finance, or autonomous systems—machine learning models are increasingly deployed to assist in decision making. Consider a healthcare application, where a predictive model is suggesting a serious condition based on a patient data. Underestimating the model's predictive uncertainty could lead to recommending a high-risk treatment that results in catastrophic outcomes. Overestimating the uncertainty, on the other hand, could lead to overly conservative decisions that forgo potentially life-saving interventions. This illustrates a fundamental trade-off in practice: balancing safety, by guarding against low-utility outcomes, with utility, by leveraging predictive information to maximize gains.

Conformal prediction (CP) has emerged as a central framework for uncertainty quantification in supervised learning, offering distribution-free, model-agnostic guarantees with efficient post-hoc implementations. In the standard setup, let $(X, Y) \sim \mathcal{D}$ be a random pair with feature $X \in \mathcal{X}$ and label $Y \in \mathcal{Y}$. CP constructs prediction sets $C(X) \subseteq \mathcal{Y}$ that satisfy a marginal coverage guarantee: $\mathbb{P}(Y \in C(X)) \geq 1-\alpha$. Despite its desirable statistical properties, it remains less clear how such sets should guide principled downstream decision making. Recent progress by Kiyani et al. (2025b) has advanced this connection by studying *risk-averse decision makers*—agents who, upon observing $X$ but not $Y$, must choose an action $a \in \mathcal{A}$ to maximize a utility function $u : \mathcal{A} \times \mathcal{Y} \to \mathbb{R}$, while ensuring that low-utility outcomes occur with probability at most $\alpha$. Specifically, given an action policy $a : \mathcal{X} \to \mathcal{A}$ and a utility certificate $\nu : \mathcal{X} \to \mathbb{R}$, they consider the problem of *maximizing the expected certificate* $\mathbb{E}[\nu(X)]$ subject to the *safety constraint*,

$$\mathbb{P}(u(a(X), Y) \geq \nu(X)) \geq 1 - \alpha. \tag{1}$$

This is a *marginal safety guarantee*, meaning the agent is safe on average over the distribution of covariates—for example, on average across patients in a medical setting. Kiyani et al. (2025b) show that the optimal solution to this problem is achieved by first constructing a prediction set $C(X)$ with valid coverage, and then acting according to the max-min decision rule: $a(X) = \arg\max_{a \in \mathcal{A}} \min_{y \in C(X)} u(a, y)$. This result establishes that conformal prediction sets are a sufficient representation of uncertainty for risk-averse agents seeking a marginal notion of safety (1).

While marginal safety offers a useful baseline, *it fails to guarantee that any particular decision is safe*—it only ensures that, on average across the population, the agent avoids low-utility outcomes. In safety-critical settings, this average guarantee can be dangerously misleading: a high-risk action might appear acceptable because its risks are averaged out by safer decisions elsewhere. From the perspective of a downstream decision maker, what matters is not marginal

[*]Equal contribution. [1]University of Pennsylvania. Correspondence to: Zihan Zhu <zhzhu1@wharton.upenn.edu>, Shayan Kiyani <shayank@seas.upenn.edu>, George Pappas <pappasg@seas.upenn.edu>, Hamed Hassani <hassani@seas.upenn.edu>.

*Proceedings of the 43rd International Conference on Machine Learning*, Seoul, South Korea. PMLR 306, 2026. Copyright 2026 by the author(s).

safety, but whether each potential action carries a reliable safety guarantee. For instance, in medical treatment planning, a decision policy may appear safe on average but fail to provide safety when high-risk actions like surgery are chosen, making action-conditional guarantees essential. This motivates the need for *action-conditional safety*: a stronger requirement that ensures the decided action will lead to high utility with high probability, conditioned on that action being taken. The action-conditional guarantees are practically meaningful and essential for safe deployment in domains such as healthcare, where decisions must be justified not globally, but per-action. Throughout the paper, we focus on finite, discrete action spaces. This is the regime in which our finite-sample theory and algorithm are stated: AC-RAC calibrates an action-specific threshold vector $\boldsymbol{\lambda} = (\lambda_a)_{a \in \mathcal{A}} \in \mathbb{R}_+^{|\mathcal{A}|}$. This scope matches many of the decision-support settings motivating our work, such as diagnostic and treatment choices, while extensions to continuous or combinatorial action spaces require additional approximation or complexity-control arguments; we return to this point in Section 6.

Formally, for an action policy $a : \mathcal{X} \to \mathcal{A}$, a utility function $u : \mathcal{A} \times \mathcal{Y} \to \mathbb{R}$, and a utility certificate $\nu : \mathcal{X} \to \mathbb{R}$, we introduce the action-conditional safety constraint: $\forall a \in \mathcal{A}$,

$$\mathbb{P}(u(a(X), Y) \geq \nu(X) \mid a(X) = a) \geq 1 - \alpha. \quad (2)$$

This constraint ensures that each action the agent may take leads, with high probability, to utility exceeding the certificate $\nu(X)$ (larger certificates are desirable). It also draws a conceptual parallel to the *multi-calibration* literature (Foster & Vohra, 1997; Zhao et al., 2021; Noarov et al., 2023; Kiyani et al., 2025a)—particularly *decision calibration*—which calibrates forecasts conditioned on the actions of downstream agents. However, such frameworks are tailored to *risk-neutral* decision makers who aim to maximize expected utility, for whom calibrated probability estimates are the correct notion of uncertainty quantification (Foster & Vohra, 1997; Zhao et al., 2021; Noarov et al., 2023; Kiyani et al., 2025a). In contrast, the *risk-averse* agents we study seek to avoid low-utility outcomes by optimizing quantiles of their utility distribution, and thus require *high-probability guarantees*. Our work can be viewed as an analogous foundation to decision calibration, adapted to the risk-averse setting.

In this work, we study such risk-averse agents who aim to maximize their average utility certificate $\mathbb{E}[\nu(X)]$, subject to satisfying the action-conditional safety guarantee in (2). This corresponds to agents optimizing their *action-conditional value-at-risk*—a formulation that captures the need for reliable utility under each possible decision. We will show that this objective admits a sharp characterization: the optimal decision policy can be implemented via conformal prediction sets that satisfy an *action-conditional*

*validity* property: for all $a \in \mathcal{A}$,

$$\mathbb{P}\left(Y \in C(X) \mid \arg\max_{a \in \mathcal{A}} \min_{y \in C(x)} u(a, y) = a\right) \geq 1 - \alpha. \quad (3)$$

That is, prediction sets with the right conditional coverage structure serve as surrogates for action-conditional risk-averse decision making. Building on this line of thinking, we now detail our contributions to both sides of the story: decision making and conformal prediction.

**On the decision making side.** We prove that prediction sets with the right conditional coverage structure serve as surrogates for action-conditional risk-averse decision making. We establish a one-sided correspondence: optimizing action-conditional prediction sets yields feasible risk-averse policies. This strengthens the results of (Kiyani et al., 2025b) and further solidifies the role of conformal prediction as a principled uncertainty quantification tool—even when conditional safety guarantees are required in downstream decisions. Moreover, we explicitly derive the *optimal prediction sets* that achieve the best population-level performance under action-conditional constraints.

**On the conformal prediction side.** To translate this population-level characterization into a practical algorithm, we develop a novel *finite-sample debiasing method* that ensures distribution-free action-conditional coverage guarantees—marking the first such result in the literature. Our calibration procedure builds on and extends the methodology of (Gibbs et al., 2025), by defining the novel notion of action specific non-conformity score. We also establish both *distribution-free lower and upper bounds* for the action-conditional coverage of our method.

**Experimental Validation.** We empirically evaluate AC-RAC on real-world tasks such as medical diagnosis. Across all settings, AC-RAC achieves significantly lower miscoverage for each action class compared to baseline methods, while maintaining competitive utility. These results highlight the practical relevance of action-conditional safety in decision-sensitive environments. Code available at `https://github.com/Telvc/AC-RAC`.

### 1.1. Related Work

**Conformal prediction.** The formal framework of conformal prediction (CP) was introduced by Vovk et al. (1999); Saunders et al. (1999); Vovk et al. (2005). CP has since become a widely adopted method for constructing prediction sets with finite-sample marginal validity guarantees (for instance look at Shafer & Vovk (2008); Angelopoulos & Bates (2023)). A large body of work has sought to strengthen the marginal guarantees of CP. These include group-conditional coverage (Barber et al., 2023; Cauchois et al., 2021; Gibbs et al., 2025; Jung et al., 2022; Vovk et al., 2005), local-

ized guarantees (Guan, 2023; Hore & Barber, 2023), label-conditional coverage (Ding et al., 2023; Vovk et al., 2005), selection-conditional coverage (Bao et al., 2024; Jin & Ren, 2025; Gazin et al., 2025; Jin & Candès, 2023), approaches for CP under treatment-based selection in counterfactual inference (Lei & Candès, 2021; Yin et al., 2024; Jin et al., 2023), robustness against covariate shifts (Tibshirani et al., 2019; Joshi et al., 2026), and set size efficiency (Sadinle et al., 2019; Kiyani et al., 2024).

**Decision calibration and multi-calibration.** Calibrated probabilistic forecasts offer an alternative to conformal prediction (CP) for uncertainty quantification, particularly suited for risk-neutral decision-makers aiming to optimize expected utility. In this context, calibration ensures that predicted probabilities align with observed frequencies, enabling rational decisions under uncertainty. Recent work has introduced *decision calibration*, which requires predictions to be indistinguishable from true outcomes for a collection of downstream decision rules (Foster & Vohra, 1998; Kakade & Foster, 2004; Zhao et al., 2021; Roth & Shi, 2024; Collina et al., 2024; Fisch et al., 2022; Kiyani et al., 2025a). This line of work emphasizes minimizing regret across all bounded-complexity agents and has led to practical recalibration procedures for multiclass settings. In parallel, the literature on *multi-calibration* has sought to guarantee calibrated predictions across overlapping subgroups, with the goal of achieving fairness and robustness (Zhao et al., 2021; Garg et al., 2024; Globus-Harris et al., 2023; Noarov & Roth, 2023; Jung et al., 2021). Our work can be viewed as a risk-averse analogue of this literature: while calibration provides guarantees for expected-utility-maximizing agents, we focus on risk-averse agents by leveraging conformal prediction to construct prediction sets that yield certified utility lower bounds for all actions.

**Risk Control Methods.** A growing body of research extends conformal prediction (CP) to control broader classes of risk measures beyond coverage guarantees (Lindemann et al., 2023; Angelopoulos et al., 2022; 2021; Cortes-Gomez et al., 2025; Lekeufack et al., 2024; Zecchin et al., 2024; Blot et al., 2024; Zecchin & Simeone, 2024). For instance, Angelopoulos et al. (2022) develop methods for conformal risk control tailored to monotone risk measures, while Cortes-Gomez et al. (2025) build prediction sets that simultaneously ensure coverage and manage risk. However, these approaches typically do not explicitly condition risk measures on the *actions* selected, nor do they investigate how prediction sets can optimally inform action-specific decision-making processes. Similarly, while Lindemann et al. (2023) and Lekeufack et al. (2024) apply CP within predefined, low-dimensional action policy spaces, they do not address the joint optimization of policy design and uncertainty quantification necessary for action-conditional guar-

antees. Exploring the combination of their setups with our action-conditional frameworks could thus lead to significantly stronger risk guarantees—a promising direction for future research.

Our work introduces a new notion: action-conditional coverage, which is fundamentally distinct. While action-conditional guarantees share similarities with group-conditional CP—specifically, in that they target coverage over groups shaped by the action policy—the key difference is that these groups are not predefined. They emerge dynamically from the decision policy, which itself depends on the prediction sets. This interdependence creates a feedback loop that breaks the usual predefine-then-calibrate pipeline and necessitates new algorithmic tools to ensure valid coverage across all actions. Similarly, selection-conditional methods address inference after a selection event in test time (e.g., coverage conditioned on the test points with small set size). However, our setting is richer: we require simultaneous coverage across all actions, not just for selected instances. Even in the binary action case (e.g., select vs. not-select), selection-conditional methods typically provide guarantees only for selected points, whereas we demand coverage for both selected and non-selected cases. Also beyond our contribution to conditional methods for debiasing sets, we also show that action-conditional prediction sets act as surrogates for risk-averse agents optimizing action-conditional value at risk, linking conditional CP with finer grained safety guarantees for decision making. For further discussion on related works, particularly on the risk averse decision making side of our contributions please see Appendix A.

## 2. Action-Conditional Conformal Risk Averse Decision Making

In this section, we will formulate the problem of risk-averse decision-making with action-conditional guarantees. We are given a space of features $\mathcal{X}$, a set of labels $\mathcal{Y}$, and a set of possible actions $\mathcal{A}$. We assume that $(x, y) \in \mathcal{X} \times \mathcal{Y}$ is drawn from an unknown joint distribution $\mathcal{D}$. After observing a feature $x$, the decision maker selects an action $a$. Note that the decision maker does not observe the true label $y$. However, the utility of the decision maker will depend on both the chosen action $a$ and the label $y$, and is captured by a given utility function $u : \mathcal{A} \times \mathcal{Y} \to \mathbb{R}$.

The core principle behind risk-averse optimization is to prioritize reliability over average performance. That is, rather than selecting actions that maximize expected utility, the agent prefers actions that avoid catastrophic low-utility outcomes with high probability. This is particularly important in safety-critical settings such as healthcare or finance, where occasional failures can be unacceptable. In the classical risk-averse setting, the agent aims to guarantee that the

utility $u(a, Y)$ is high with probability at least $1 - \alpha$. For a pre-specified risk tolerance level $\alpha \in (0, 1)$, the $\alpha$-level quantile of the utility for each action $a \in \mathcal{A}$ is defined as

$$\nu_\alpha(a; x) = \text{quantile}_\alpha\left(u(a, Y) \mid X = x\right).$$

A risk-averse decision maker then selects the action that maximizes this guaranteed utility level—that is, the action whose worst-case performance, up to level $1 - \alpha$, is as high as possible. This leads to conservative decisions that avoid low-utility outcomes with high confidence.

This formulation requires full knowledge of the conditional distribution $p(y \mid x)$, which is typically inaccessible. To address this, (Kiyani et al., 2025b) introduce a marginal relaxation called RA-DPO (Risk-Averse Decision Policy Optimization), where the quantile constraint is enforced over the entire data distribution:

$$\max_{a(\cdot), \nu(\cdot)} \quad \mathbb{E}_X[\nu(X)], \tag{4}$$
$$\text{s.t.} \quad \mathbb{P}(u(a(X), Y) \geq \nu(X)) \geq 1 - \alpha,$$

where $a : \mathcal{X} \to \mathcal{A}$ is the policy and $\nu : \mathcal{X} \to \mathbb{R}$ is the associated utility certificate. This marginal formulation provides a statistically feasible proxy for pointwise quantile guarantees.

In this paper, we propose a stronger notion of risk-aversion: one that is *action-conditional*. That is, the utility guarantee must hold within each subpopulation of the data defined by the action taken. This is essential in applications where actions have distinct operational meanings (e.g., treatment decisions), and fairness or reliability must be certified at the level of each action.

Formally, we impose the requirement that for each action $a \in \mathcal{A}$, the agent achieves a high-utility outcome with probability at least $1 - \alpha$ conditional on selecting that action. This leads to the following optimization problem, termed as *Action-Conditional Decision Policy Optimization* (AC-DPO):

$$\max_{a(\cdot), \nu(\cdot)} \quad \mathbb{E}_X[\nu(X)],$$
$$\text{s.t.} \quad \mathbb{P}(u(a(X), Y) \geq \nu(X) \mid a(X) = a) \geq 1 - \alpha, \tag{5}$$

where the constraint holds **for all** $a \in \mathcal{A}$. Here, recall that $u$ denotes the utility function, $a : \mathcal{X} \to \mathcal{A}$ is the decision policy, $\nu : \mathcal{X} \to \mathbb{R}$ is the quantile function, and $\alpha$ is the risk tolerance level. This formulation strengthens the guarantee in (4) by ensuring calibrated performance across all decision branches, not only in aggregate. We refer to (5) as the action-conditional counterpart to RA-DPO.

## 2.1. Constructing Action-Conditional Policies from Prediction Sets

Prediction sets have emerged as a central tool in uncertainty quantification due to their ability to provide distribution-free, model-agnostic coverage guarantees. Recent work by Kiyani et al. (2025b) has established a fundamental connection between prediction sets and risk-averse decision making. In particular, it is shown that that any optimal decision policy under marginal safety constraints can be equivalently realized through a max-min decision rule over appropriately constructed prediction sets. Formally, the solution of RA-DPO (4) is equivalent to the solution of a prediction-set-based optimization problem called Risk-Averse Conformal Prediction Optimization (RA-CPO),

$$\max_{C(\cdot)} \quad \mathbb{E}_X\left[\max_{a \in \mathcal{A}} \min_{y \in C(X)} u(a, y)\right]$$
$$\text{s.t.} \quad \mathbb{P}(Y \in C(X)) \geq 1 - \alpha.$$

Using an optimal solution $C(x)$ of this problem, we can find the optimal policy for (4) as

$$a_{\text{RA}}^C(x) = \arg\max_{a \in \mathcal{A}} \min_{y \in C(x)} u(a, y),$$
$$\nu_{\text{RA}}^C(x) = \max_{a \in \mathcal{A}} \min_{y \in C(x)} u(a, y). \tag{6}$$

This equivalence shows that prediction sets communicate uncertainty effectively and serve as a complete representation for optimizing risk-averse utility under marginal coverage constraints.

We now generalize this result to the *action-conditional* setting, where safety must hold *conditionally* on each action being taken. Specifically, we consider the *Action-Conditional Risk-Averse Conformal Prediction Optimization* (AC-CPO):

$$\max_{C(\cdot)} \quad \mathbb{E}_X\left[\max_{a \in \mathcal{A}} \min_{y \in C(X)} u(a, y)\right]$$
$$\text{s.t.} \quad \mathbb{P}\left(Y \in C(X) \mid a_{\text{RA}}^C(X) = a\right) \geq 1 - \alpha, \quad \forall a \in \mathcal{A}. \tag{7}$$

We next show that this action-conditional generalization leads to a feasible policy optimization formulation (defined in (5)). That is, prediction sets constitute a relaxed representation for risk-averse decision making when conditioning on individual actions.

**Theorem 2.1** (From AC-CPO to AC-DPO). *From any optimal solution $C^*$ to (7), we can construct a feasible solution $(a_{\text{RA}}^{C^*}(x), \nu_{\text{RA}}^{C^*}(x))$ for (5) such that it follows the action-conditional constraints and*

$$\mathbb{E}_X\left[\nu_{\text{RA}}^{C^*}(X)\right] = \mathbb{E}_X\left[\max_{a \in \mathcal{A}} \min_{y \in C^*(X)} u(a, y)\right].$$

We refer the reader to Appendix C.1 for the proof. In contrast to the two-way equivalence shown by Kiyani et al.

(2025b), our action-conditional setting yields only a one-directional guarantee: an optimal AC-CPO solution induces a feasible AC-DPO policy with equal value, whereas the converse need not hold. This asymmetry arises because, under action-conditional constraints, the conditioning event in AC-CPO $\{a_{\text{RA}}^C(X) = a\}$ is endogenous to the prediction sets but the event in AC-DPO $\{a(X) = a\}$ is free. Despite this relaxation, our experiments (Section 5) show that the resulting policies preserve high utility while delivering valid action-conditional safety. Accordingly, we use AC-CPO as a principled surrogate that preserves conditional guarantees and provides a constructive route to implementable policies. A simple example illustrates where the surrogate can be conservative. Suppose a safe default action is moderately good across all labels, whereas a specialized action is optimal only on a small subset of easy inputs and performs poorly elsewhere. A free AC-DPO policy can select the specialized action only on those easy inputs. In contrast, AC-CPO must certify coverage on the subpopulation induced by the conformal set and the max-min rule itself, so points near the boundary between the specialized and default actions may force larger sets or fallback to the default action. This is the precise feedback loop that makes the action-conditional problem harder than the marginal case.

## 3. Reparameterized Construction of Prediction Sets

In this section, we study the population-level conformal prediction problem with action-conditional guarantees, as defined in (7). Our goal is to characterize the prediction set function $C(x)$ that maximizes risk-averse utility while satisfying per-action coverage constraints. To this end, we introduce a reparameterization and derive a dual formulation using tools from convex duality. See Theorem 3.2 for the final characterization. The principles developed here will serve as the foundation for designing a finite-sample method in the next section. All technical proofs are deferred to Appendix C.2. We begin by defining two quantities that encapsulate the risk-averse utility structure:

$$
\begin{aligned}
\theta(x,t) &= \max_{a \in \mathcal{A}} \text{quantile}_{1-t}\left[u(a,Y) \mid X = x\right], \\
a(x,t) &= \arg\max_{a \in \mathcal{A}} \text{quantile}_{1-t}\left[u(a,Y) \mid X = x\right].
\end{aligned}
\tag{8}
$$

Here, $\theta(x,t)$ denotes the maximum achievable risk-averse utility at coverage level $t \in (0,1)$, and $a(x,t)$ denotes the action that attains this utility. Intuitively, we aim to assign a feature-dependent coverage threshold $t(x) \in [0,1]$, such that the resulting prediction set $C(x)$ satisfies action-conditional coverage and induces high utility. To this end, we reparameterize the prediction set problem (7) in terms of the pointwise coverage function $t(x) = \mathbb{P}(Y \in C(X) \mid X = x)$. We show that (7) can be relaxed to the following

optimization problem over coverage functions:

$$
\boxed{
\begin{aligned}
\max_{t:\mathcal{X}\to[0,1]} \quad & \mathbb{E}_X\left[\theta(X, t(X))\right], \\
\text{s.t.} \quad & \mathbb{E}\left[t(X) \mid a(X, t(X)) = a\right] \geq 1 - \alpha, \forall a \in \mathcal{A}.
\end{aligned}
}
\tag{9}
$$

The next proposition makes this relaxation precise and gives a closed-form expression for the prediction sets:

**Proposition 3.1** (Set Characterization). *From any optimal solution $t^*(x)$ to (9), we obtain a feasible coverage function $C^*(x)$ for (7) such that*

$$
\mathbb{E}_X\left[\max_{a \in \mathcal{A}} \min_{y \in C^*(X)} u(a,y)\right] = \mathbb{E}_X\left[\theta(X, t^*(X))\right].
$$

*The prediction set and decision policy are given by:*

$$
\begin{aligned}
C^*(x) &= \{y \in \mathcal{Y} : u(a(x, t^*(x)), y) \geq \theta(x, t^*(x))\}, \\
a^*(x) &= a(x, t^*(x)).
\end{aligned}
\tag{10}
$$

*Moreover, this prediction set satisfies $t^*(x) = \mathbb{P}(Y \in C^*(X) \mid X = x)$.*

This reparameterization serves as a conceptual bridge between the prediction set formulation and the risk-averse decision policy. Once the optimal function $t^*(x)$ is obtained, we can construct both the feasible prediction sets and the associated actions via (10). We now turn to solving (9). Unlike the marginal setting in Kiyani et al. (2025b), where the coverage constraint is convex and decoupled from the decision rule, the action-conditional formulation introduces new complexity: the conditioning event $a(X, t(X)) = a$ depends nonlinearly on the decision variable $t$. This dependence breaks convexity and prevents direct application of previous techniques. We begin by rewriting the action-conditional constraint in (9) using the identity

$$
\mathbb{E}_X[t(X)\mathbb{1}_a(X, t(X))] \geq (1-\alpha)\mathbb{P}(a(X, t(X)) = a),
$$

where $\mathbb{1}_a(x,t) = \mathbb{1}[a(x,t) = a]$ denotes the indicator that action $a$ is selected under threshold $t$. This reformulation casts both sides of the constraint as expectations, enabling a more linear treatment of the dependence on $t$ and the decision rule $a(x,t)$.

A key challenge, however, is that the indicator function $\mathbb{1}_a(x,t)$ introduces a discontinuous dependence on $t$, leading to nonconvexity of the problem. We handle this difficulty by developing new techniques detailed in Appendix C.2.2, drawing on tools from duality theory. To proceed, we define the threshold function $t^*(x, \boldsymbol{\lambda})$ using a family of parameters $\boldsymbol{\lambda} = (\lambda_a)_{a \in \mathcal{A}} \in \mathbb{R}_{\geq 0}^{|\mathcal{A}|}$, one for each action

$$
t^*(x, \boldsymbol{\lambda}) =
$$

$$
\arg\max_{t \in [0,1]} \left\{\theta(x,t) + \sum_{a \in \mathcal{A}} \lambda_a(t - (1-\alpha))\mathbb{1}_a(x,t)\right\}.
\tag{11}
$$

This expression captures the tradeoff between utility and action-conditional coverage through a tunable vector of $|\mathcal{A}|$-dimensional parameters $\boldsymbol{\lambda}$. The next theorem proves that this formulation can fully describe the optimal solution to the problem (9).

**Theorem 3.2** (Strong Duality). *Assume the marginal distribution $\mathcal{P}_X$ is continuous. Then strong duality holds for the reparameterized problem (9). In particular, there exists $\boldsymbol{\lambda}^* \in \mathbb{R}_{\geq 0}^{|\mathcal{A}|}$ such that*

$$t_{\mathrm{opt}}(x) = t^*(x, \boldsymbol{\lambda}^*)$$

*is the optimal solution to (9). Moreover, the dual minimizer is given by:*

$$\boldsymbol{\lambda}^* = \arg\min_{\boldsymbol{\lambda} \in \mathbb{R}_{\geq 0}^{|\mathcal{A}|}} \Psi(\boldsymbol{\lambda}), \quad \text{where} \quad \Psi(\boldsymbol{\lambda}) =$$

$$\mathbb{E}_X \left[ \max_{t \in [0,1]} \left\{ \theta(X, t) + \sum_{a \in \mathcal{A}} \lambda_a(t - (1 - \alpha)) \mathbb{1}_a(X, t) \right\} \right]$$

This result provides a tractable characterization of problem (9) with action-dependent constraints.

## 4. Action-Conditional Risk Averse Calibration

In this section, we consider the risk-averse optimization problem in the finite-sample setting. Suppose that we have access to a set of calibration samples $\{(x_i, y_i)\}_{i \in [n]}$ and a predictive model $f : \mathcal{X} \to \Delta(\mathcal{Y})$, which assigns each $x \in \mathcal{X}$ to a probability vector in $\Delta(\mathcal{Y})$. We denote the output of $x$ as $f_x$, which is an approximation of the distribution of the label $y$ given the input $x$. In practice, $f$ is the (softmax) output of a pre-trained model that predicts label $y$ from $x$. In this section, we aim to develop a finite sample algorithm that connects predictions to actions that can exploit any black-box pre-trained predictive model. We present all the technical proofs in Appendix C.3.

Given the model $f$, we estimate functions $\theta$ and $a$ defined in (8), via replacing the true conditional probabilities with their estimated counterparts obtained by $f$. In more detail, we obtain $\hat{\theta}$ and $\hat{a}$ via

$$\hat{\theta}(x, t) = \max_{a \in \mathcal{A}} \text{quantile}_{1-t} \big[ u(a, Y) \mid Y \sim f_x \big],$$
$$\hat{a}(x, t) = \arg\max_{a \in \mathcal{A}} \text{quantile}_{1-t} \big[ u(a, Y) \mid Y \sim f_x \big]. \quad (12)$$

For any $\boldsymbol{\lambda} \in \mathbb{R}_{\geq 0}^{|\mathcal{A}|}$, define the optimal assignment function

$$\hat{t}(x, \boldsymbol{\lambda}) = \arg\max_t \left( \hat{\theta}(x, t) + \sum_a \lambda_a(t - (1 - \alpha)) \mathbb{1}_{\hat{a}(x,t)=a} \right) \quad (13)$$

and the conformal prediction set $\hat{C}(x, \boldsymbol{\lambda}) = \{y \in \mathcal{Y} : u(\hat{a}(x, \hat{t}(x, \boldsymbol{\lambda})), y) \geq \hat{\theta}(x, \hat{t}(x, \boldsymbol{\lambda}))\}$.

We note that all the functions can be easily computed because they are parametrized by $\boldsymbol{\lambda}$. From Theorem 3.2 we know that there exists a $\boldsymbol{\lambda}^*$ such that the optimal prediction sets is derived using the function $\hat{t}(x, \boldsymbol{\lambda}^*)$ and $\hat{C}(x, \boldsymbol{\lambda}^*)$. Thus, the next question is now to select a feasible $\boldsymbol{\lambda}^*$ that satisfies the action-conditional coverage constraints. To do this, we first reparametrize the event $\{Y \in \hat{C}(x, \boldsymbol{\lambda})\}$ by a closed-form of $\boldsymbol{\lambda}$. In more detail, we note that $\hat{C}$ is determined by the assignment function $\hat{t}$, which can be decoupled by its one-dimensional counterpart. For any $a \in \mathcal{A}$, define $\hat{t}(x, \lambda_a)$ as

$$\hat{t}(x, \lambda_a) = \arg\max_{t \in [0,1]} \big( \hat{\theta}(x, t) + \lambda_a(t - (1 - \alpha)) \big),$$

then $\hat{t}(x, \lambda_a)$ is only parametrized by the scalar $\lambda_a$. We further define the action specific nonconformity score $\lambda_a^*$ for any $(x, y)$ as follows

$$\lambda_a^*(x, y) = \inf \Big\{ \lambda_a \geq 0 : y \in$$
$$\text{QuantileSet}_{1-\hat{t}(x, \lambda_a)} [u(a, Y) \mid Y \sim f_x] \Big\}.$$

where the quantile set is defined as

$$\text{QuantileSet}_{1-t} [u(a, Y) \mid Y \sim f_x]$$
$$= \{y \in \mathcal{Y} : \mathbb{P}_{Y \sim f_x}(u(a, Y) \leq u(a, y)) \leq 1 - t\}. \quad (14)$$

Intuitively, the larger the value of $\lambda_a^*(x, y)$, the more conservative it is to select action $a$ at input $x$ when facing outcome $y$. This quantifies how atypical $y$ is for $x$, if we want to play action $a$. The lemma below, under mild tie-breaking assumptions, describes an important property of these scores.

**Lemma 4.1.** *Conditioning on the event $\{\hat{a}(x, \hat{t}(x, \boldsymbol{\lambda})) = a\}$, the following events are equivalent*

$$\{Y \in \hat{C}(x, \boldsymbol{\lambda})\} = \{\lambda_a \geq \lambda_a^*(x, Y)\}.$$

Lemma 4.1 implies membership in the set $\hat{C}(x, \boldsymbol{\lambda})$ reduces to the scalar threshold test $\{\lambda_a \geq \lambda_a^*(x, Y)\}$ conditioned on picking action $a$. In other words, for each $(x, y)$, the action specific nonconformity score $\lambda_a^*(x, y)$, is the value such that $y$ lies in the prediction set exactly when $\lambda_a \geq \lambda_a^*(x, y)$.

To calibrate the $|\mathcal{A}|$-dimensional vector $\boldsymbol{\lambda}$, we propose a novel formulation based on pinball loss minimization using action specific nonconformity scores. Specifically, we include an imputed test pair $(x_{\text{test}}, y)$ alongside the $n$ calibration samples $\{(x_i, y_i)\}_{i=1}^n$, resulting in $n + 1$ total points. For each sample $i = 1, \ldots, n$ and the test point, we define

the loss:

$$F_y(\boldsymbol{\lambda}) = \frac{1}{n+1} \sum_{i=1}^{n} \ell_\alpha \Big( \sum_a \lambda_a \, \mathbb{1}_{\{\hat{a}(x_i, \hat{t}(x_i, \boldsymbol{\lambda})) = a\}},$$

$$\sum_a \lambda_a^*(x_i, y_i) \, \mathbb{1}_{\{\hat{a}(x_i, \hat{t}(x_i, \boldsymbol{\lambda})) = a\}} \Big)$$

$$+ \frac{1}{n+1} \ell_\alpha \Big( \sum_a \lambda_a \, \mathbb{1}_{\{\hat{a}(x_{\text{test}}, \hat{t}(x_{\text{test}}, \boldsymbol{\lambda})) = a\}},$$

$$\sum_a \lambda_a^*(x_{\text{test}}, y) \, \mathbb{1}_{\{\hat{a}(x_{\text{test}}, \hat{t}(x_{\text{test}}, \boldsymbol{\lambda})) = a\}} \Big),$$

where the pinball loss is defined as $\ell_\alpha(u, v) = (v - u)(\mathbb{1}_{u \le v} - \alpha)$. Minimizing $F_y(\boldsymbol{\lambda})$ over $\boldsymbol{\lambda} \in \mathbb{R}_{\ge 0}^{|\mathcal{A}|}$ thus yields a set of parameters that encode action-specific thresholds, leading to finite-sample coverage guarantees. We can now present our main finite sample algorithm in 1.

---

**Algorithm 1** Action-Conditional Risk Averse Calibration (AC-RAC)

---

**Input:** Miscoverage level $\alpha$, calibration samples $\{(x_i, y_i)\}_{i=1}^n$, test covariate $x_{\text{test}}$

**for** *each* $y \in \mathcal{Y}$ **do**

$\quad \Big\lfloor \quad \hat{\boldsymbol{\lambda}}_y = \arg\min_{\boldsymbol{\lambda} \in \mathbb{R}_{\ge 0}^{|\mathcal{A}|}} F_y(\boldsymbol{\lambda}).$

**Output:** $C_{\text{final}}(x_{\text{test}}) = \{y \in \mathcal{Y} \mid y \in \hat{C}(x_{\text{test}}, \hat{\boldsymbol{\lambda}}_y)\}$

---

*Remark* 4.2. We optimize $F_y(\boldsymbol{\lambda})$ using projected subgradient descent, since the objective is suitable for first-order optimization but has no closed-form minimizer in general. By Lemma 4.1, after conditioning on the induced action, each sample contributes through the scalar comparison $\lambda_a \ge \lambda_a^*(x, y)$. Thus each iteration reduces to computing the induced actions and the corresponding action-wise empirical coverage terms. We present the full algorithmic details and computational-efficiency analysis in Appendix B.1, and provide the subgradient derivation and convergence statement in Appendix C.4.

Next, we show that AC-RAC outputs prediction sets with distribution free guarantees at test time.

**Theorem 4.3** (Finite-Sample Action-Conditional Validity). *Let* $(x_1, y_1), \ldots, (x_n, y_n), (x_{\text{test}}, y_{\text{test}})$ *be exchangeable samples. Then for every* $a \in \mathcal{A}$,

$$\mathbb{P}\big(y_{\text{test}} \in C_{\text{final}}(x_{\text{test}}) \mid a_{\text{RA}}^{C_{\text{final}}}(x_{\text{test}}) = a\big) \ge 1 - \alpha.$$

*If* $(x_i, y_i)$ *are i.i.d. and the score* $\lambda_a^*(X, Y)$ *has a continuous distribution for all* $a$, *then*

$$\mathbb{P}\big(y_{\text{test}} \in C_{\text{final}}(x_{\text{test}}) \mid a_{\text{RA}}^{C_{\text{final}}}(x_{\text{test}}) = a\big) \le$$

$$1 - \alpha + \frac{|\mathcal{A}|}{(n+1) \cdot \mathbb{P}(a_{\text{RA}}^{C_{\text{final}}}(x_{\text{test}}) = a)} \quad \text{for each } a \in \mathcal{A}.$$

Theorem 4.3 guarantees that the proposed method achieves valid action-conditional coverage with finite samples and

the coverage error is tightly controlled and decays at a rate of $1/n$. The i.i.d. and continuity condition is purely a tie-breaking device: it guarantees that, with probability 1, at most one calibration point per action lands exactly on the set boundary, which we need to bound the slack term in the upper–tail inequality. Such tie-breaking assumptions are standard in finite-sample conformal-prediction bounds and can always be enforced in practice by adding an arbitrarily small random jitter to the conformity scores, without affecting coverage or utility. We remark that our proof steps are inspired by Gibbs et al. (2025) and the results are analogous (see Theorem 3 in Gibbs et al. (2025)).

**Corollary 4.4.** *Assume that* $(x_1, y_1), \ldots, (x_n, y_n),$ $(x_{\text{test}}, y_{\text{test}})$ *are exchangeable. We then have*

$$\mathbb{P}\Big( u\left(a_{\text{RA}}^{C_{\text{final}}}(x_{\text{test}}), y_{\text{test}}\right) \ge \nu_{\text{RA}}^{C_{\text{final}}}(x_{\text{test}}) \mid$$

$$a_{\text{RA}}^{C_{\text{final}}}(x_{\text{test}}) = a \Big) \ge 1 - \alpha, \quad \forall a \in \mathcal{A}.$$

Corollary 4.4 ensures that the max-min decision policy applied to the final prediction set $C_{\text{final}}$ yields a distribution-free, action-conditional safety guarantee. Specifically, the utility achieved by the selected action exceeds the certified risk-averse threshold with probability at least $1 - \alpha$, conditional on the action taken. Combined with Theorem 2.1, this result confirms that the output of our finite-sample algorithm satisfies the original action-conditional risk-averse objective in (7), thereby closing the loop between population-level theory and implementable practice.

## 5. Numerical Experiments

In this section, we empirically evaluate the performance of the proposed action-conditional algorithm (denoted by AC-RAC) on two experimental setups. We begin by describing the competing baselines, followed by the evaluation metrics. We compare AC-RAC against two families of methods, each instantiated with the same pre-trained probabilistic model $f \colon (x, y) \mapsto f_x(y)$, where $f_x(y)$ is the assigned probability of the input-label pair $(x, y)$.

**Baseline 1: Conformal Prediction with Max-Min Decision Rule.** We benchmark our method against established conformal inference techniques by generating $(1-\alpha)$-valid prediction sets $C(x)$ using split conformal prediction. Among these, we include the `RAC` method of (Kiyani et al., 2025b), which represents the state-of-the-art approach for risk-averse decision making under *marginal* safety guarantees. `RAC` employs a risk-sensitive scoring function to construct prediction sets independent of the action space, followed by a max-min decision rule that ensures robust performance against worst-case outcomes. In addition to `RAC`, we include two action-independent conformal baselines. `score-1` (Sadinle et al., 2019): inverse probability

score, defined as $1 - f_x(y)$; score-2 (Romano et al., 2020): cumulative tail mass, $\sum_{y':f_x(y')>f_x(y)} f_x(y')$. All methods yield a prediction set $C(x)$, which is subsequently paired with a max-min utility decision rule: $a_{\text{RA}}^C(x) = \arg\max_{a \in \mathcal{A}} \min_{y \in C(x)} u(a, y)$, selecting the safest action that maximizes worst-case utility across the set.

**Baseline 2: Calibrated Best-Response.** We also evaluate against a calibration-based decision-making baseline. This approach applies the decision calibration procedure of (Noarov et al., 2023) to adjust the predictive model on a held-out calibration set, ensuring improved reliability through swap regret bounds introduced in Zhao et al. (2021). Once the calibrated forecast $f_x(y)$ is obtained, actions are selected to maximize expected utility: $a_{\text{BR}}(x) = \arg\max_{a \in \mathcal{A}} \mathbb{E}_{y \sim f_x}[u(a, y)]$. This standard risk-neutral method performs well on average utility but lacks coverage guarantees and is vulnerable to high-risk errors under underestimated uncertainty.

**Evaluation Metrics.** For methods in Baseline 1, we evaluate performance using the following two metrics. **Action-specific miscoverage:** the empirical miscoverage rate conditioned on each selected action, assessing conditional reliability; **Worst-case utility:** the average realized utility under the max-min rule, $\mathbb{E}[\max_a \min_{y \in C(x)} u(a, y)]$, capturing conservative performance. In addition, to compare all methods (including Calibrated Best-Response) we report **Critical error rate**, which is the fraction of samples where the selected action leads to a highly adverse utility outcome, indicating the frequency of catastrophic decisions. To quantify the informativeness of the resulting prediction sets, we also report mean prediction-set size and the false discovery rate (FDR), defined as $\mathbb{E}[|C(X) \setminus \{Y\}|/|C(X)|]$. These diagnostics help distinguish genuine action-conditional calibration from overly conservative set enlargement. Additional evaluations, including average utility, sweeps over multiple $\alpha$ values for action-specific miscoverage, set-size diagnostics, FDR, rare-action stress tests, and action-space scaling ablations are presented in Appendix D.

### 5.1. Medical Diagnosis

We evaluate AC-RAC in the context of risk-sensitive medical diagnosis and treatment planning. Our dataset is the COVID-19 Radiography Database (Chowdhury et al., 2020; Rahman et al., 2021), comprising chest X-ray images labeled as *Normal*, *Pneumonia*, *COVID-19*, or *Lung Opacity*. Images are partitioned at random into training (70%), calibration (10%), and test (20%) subsets.

For feature extraction, we use Inception-v3 (Szegedy et al., 2015; 2016) pretrained on ImageNet, fine-tuned from the second inception block. Clinical trade-offs are encoded via a utility matrix (Table 1) mapping each diagnosis to a set of actions. While we report results using this matrix, our

framework supports alternative specifications (see Appendix B of Kiyani et al. (2025b)). All baselines are calibrated to ensure consistent mapping of model outputs to the four actions.

Figure 1 (top row) presents the results of the medical diagnosis task at a nominal miscoverage level of $\alpha = 0.05$. The left panel shows that AC-RAC is the only method achieving valid conditional coverage across all actions, while RAC, score-1, and score-2 systematically over- or under-cover. The middle panel reports the average realized max-min utility across $\alpha \in \{0.01, 0.02, 0.03, 0.05, 0.1\}$, revealing that AC-RAC achieves higher worst-case utility than score-1 and score-2, and slightly lower utility than RAC. This gap is expected, as RAC is optimized for marginal coverage, while AC-RAC imposes stricter action-conditional guarantees, which introduces a modest trade-off in utility. The right panel shows the rate of critical errors, where a high-risk action is selected for a vulnerable patient group. Here, AC-RAC reduces harmful decisions by a significant margin, highlighting its advantage in safety-critical settings like clinical decision-making. The omission of action 2 (Quarantine) for the baselines in the left panel is not a plotting artifact: RAC, score-1, score-2, and the calibrated best-response policy never select that action on this task, so their action-conditional errors are undefined for it. AC-RAC, by contrast, selects Quarantine on 2.72% of test instances and keeps this rare action in the calibration loop. Appendix D.3 further shows that the stronger guarantee does not arise from excessive conservativeness: at $\alpha = 0.05$, AC-RAC increases the mean set size relative to RAC only from 3.252 to 3.373 (about 3.7%) and has similar overall FDR (0.699 versus 0.682).

### 5.2. Recommender Systems

We next evaluate AC-RAC in the context of recommendation systems, where the goal is to decide about whether to suggest a movie to a user. Each feature corresponds to a user–item pair $x = (u, m)$, where $u$ and $m$ are feature vectors representing the user and the movie, respectively. The associated label $y \in \{1, 2, 3, 4, 5\}$ denotes the rating assigned by the user. The action space is binary: $\mathcal{A} = \{\text{No-Rec}, \text{Rec}\}$, where Rec indicates recommending the movie. We partition the dataset into three splits: 80% for training the predictive model, 10% for calibration, and 10% for evaluation.

At decision time, utility of each action depends on the true rating (specified in Table 2). Recommending a movie with rating $y$ yields utility $u(\text{Rec}, y) = y - 3$, encouraging recommendations only when the rating is above average. The no-recommendation action has utility $u(\text{No-Rec}, y) = 0$, reflecting a conservative fallback. This setup captures the trade-off between opportunity and risk: aggressive recom-

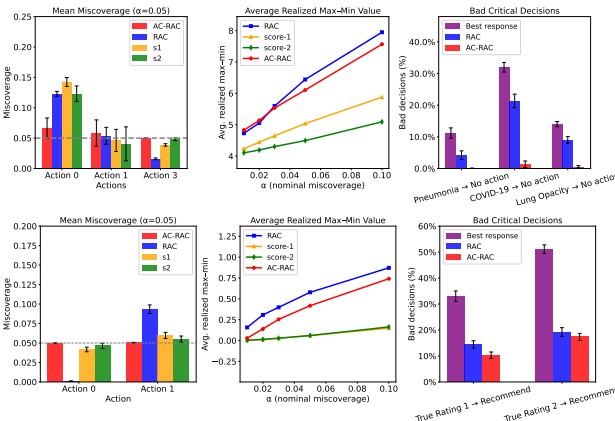

*Figure 1.* Comparison of methods under action-conditional constraints at nominal level $\alpha = 0.05$. Each row shows (left) mean miscoverage across actions, (center) average realized max-min utility, and (right) critical decision error rate. The first row is the result of medical diagnosis and second row is recommender system. All the error bars are averaged over 40 random seeds. In the medical-diagnosis miscoverage panel, baseline values for action 2 are omitted because those methods never select that action, so the corresponding conditional miscoverage is undefined.

mendations can yield high returns but carry greater downside under uncertainty.

Figure 1 (bottom row) summarizes the performance of each method on a recommendation task under $\alpha = 0.05$. The left panel confirms that AC-RAC uniquely achieves calibrated action-conditional coverage, whereas the other methods suffer from miscalibration. This misalignment between coverage and decision type leads to reliability gaps. The middle panel shows that AC-RAC achieves worst-case utility comparable to RAC, while offering significantly better action-conditional coverage. The right panel reveals that AC-RAC substantially reduces the rate of critical recommendation errors.

## 6. Conclusion and Discussion

We introduced CP sets with action-conditional guarantees, and then showed these sets provide a feasible policy for risk averse decision makers seeking action conditional safety guarantees. Building upon that we construct finite-sample CP sets with provable action-conditional validity. While this work provides new avenues for principled uncertainty quantification in safety-critical domains, it will be interesting to explore other notions of safety beyond quantiles (such as CVaR). The present theory and algorithm are deliberately scoped to finite, discrete action spaces. One natural extension is to discretize a continuous action space and then apply AC-RAC on the discretized set, which preserves the current finite-sample guarantee on that discretization. A more ambitious direction is to parameterize the action-dependent

threshold $\lambda(a)$ using linear features, kernels, or neural networks and optimize the resulting pinball objective over that function class. Such an extension would require new arguments controlling the complexity of the function class and the induced action-dependent conditioning events, so the current guarantee would not transfer verbatim.

## Impact Statement

We develop action-conditional conformal prediction methods that provide finite-sample guarantees for risk-averse decision making, aiming to improve the reliability of learning-enabled decision pipelines. While such tools may influence deployment practices in high-stakes settings (e.g., healthcare or recommendation), we do not identify any specific societal consequences that warrant separate discussion beyond the technical scope of this work.

## Acknowledgement

The authors thank EnCORE, the Institute for Emerging CORE Methods in Data Science, for their support. SK additionally acknowledges support from a gift from AWS to Penn Engineering's ASSET Center for Trustworthy AI.

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

# A. Further Related Work

**Economic Literature on Risk Aversion.** Decision-making under risk aversion is a classical topic in economics, with foundational concepts introduced by Arrow (1965), (Pratt, 2013), and studies on stochastic dominance (Hadar & Russell, 1969; Meyer, 1977). Recent economic research has increasingly focused on conditional guarantees in decision-making, emphasizing that risk assessments must reflect specific decision contexts rather than aggregate or marginal measures. For example, Steffensen & Søe (2023) develop a continuous-time decision model with state-dependent risk aversion, showing how optimal choices adjust with health or economic status. Similarly, Desmettre & Steffensen (2023) model portfolio selection under random risk preferences, highlighting the role of expectation over uncertain attitudes. Lim & Ravaioli (2021) provide experimental evidence for set-dependent preferences, where risk aversion is shaped by the choice environment, while Brandtner et al. (2020) demonstrate how shortfall risk measures allow for richer, context-aware modeling of investor risk tolerance. In parallel, Tao et al. (2025) introduce a contextual risk framework where decisions are optimized with respect to one uncertainty while conditioning on another, and Batista (2023) incorporates subgroup-specific risk aversion into macroeconomic modeling to better explain international puzzles.

However, these works predominantly rely on explicit parametric assumptions or full knowledge of underlying distributions. In contrast, our approach focuses on learning and uncertainty quantification aspects explicitly, developing distribution-free methods capable of leveraging any black-box pretrained model.

**Bayesian Approaches to Risk-Averse Decision-Making.** Bayesian methods provide another prominent framework for risk-averse decision-making, commonly utilizing models like Gaussian Processes (GPs) to quantify uncertainty and optimize measures such as Value-at-Risk (VaR). Notable examples include safe exploration in optimization settings (Sui et al., 2015), direct VaR optimization using GPs (Nguyen et al., 2021), and dose allocation strategies in precision medicine (Demirel et al., 2022). Recent works have extended Bayesian methods toward conditional risk-averse decision making by optimizing risk-sensitive metrics such as Value-at-Risk (VaR) or Conditional Value-at-Risk (CVaR) with respect to specific actions or policies. Cakmak et al. (2020) introduce a Bayesian optimization framework that directly targets risk measures over outcome distributions using Gaussian Processes, offering action-dependent safety. Lin et al. (2022) propose Bayesian Risk MDPs, which evaluate actions based on posterior CVaR, yielding action-dependent safety in sequential decision-making. Baudry et al. (2021) develop Thompson Sampling algorithms for multi-armed bandits that optimize CVaR, guaranteeing robust lower-tail performance per action. Most recently, Jia et al. (2024) introduce a Bayesian policy optimization method using chance constraints on a novel average conditional risk measure, providing conditional guarantees across subpopulations.

These approaches rely on accurate Bayesian posterior distributions, thus implicitly assuming well-specified probabilistic models. Our conformal approach complements rather than competes with Bayesian methods: our theoretical results (up to Section 3) can be directly employed even in Bayesian settings. In fact, when Bayesian approximations are reliable, one can take advantage of our optimal prediction sets derivation in population, and then calibrate the prediction sets with finite sample under Bayesian models, without employing the finite-sample calibration of Section 4. Alternatively, even when Bayesian assumptions' precision is uncertain, one can still start from Bayesian posteriors and further calibrate prediction sets using our approach, ensuring robust safety guarantees.

**Conditional robust optimization (CRO).** Robust optimization (RO) and distributionally robust optimization (DRO) are classical paradigms for decision making under uncertainty: RO optimizes against worst-case realizations over a prescribed uncertainty set, while DRO optimizes against worst-case distributions within an ambiguity set (Ben-Tal et al., 2009; Bertsimas & Sim, 2004; Delage & Ye, 2010; Rahimian & Mehrotra, 2019; Goh & Sim, 2010). A central recent direction is to incorporate *context* $x$ by learning context-dependent uncertainty/ambiguity sets $\mathcal{U}(x)$, often termed conditional robust optimization (CRO). Early data-driven RO developed tractable, data-derived uncertainty sets—e.g., via residual-based constructions or ML-guided set design—typically without explicit conditioning (Tulabandhula & Rudin, 2014; Bertsimas et al., 2018). CRO extends this by allowing the set to adapt with $x$, enabling finer protection when uncertainty varies across contexts. Recent methods build $\mathcal{U}(x)$ using local neighborhoods or clustering in covariate space, contextual parametric families, or decision-focused learning objectives that optimize downstream robust performance (Chenreddy et al., 2022; Ohmori, 2021; Wang et al., 2025). Complementary work develops end-to-end CRO by differentiating through the robust optimization layer to learn $\mathcal{U}(x)$ jointly with predictors, improving empirical robustness but typically relying on additional modeling/algorithmic assumptions beyond distribution-free guarantees (Chenreddy & Delage, 2024). Relatedly, contextual DRO and decision-dependent ambiguity sets have been explored, where uncertainty modeling depends on both covariates and decisions (Zhu et al., 2024). Recent work also connects conformal prediction to CRO by using conformal regions to

build uncertainty sets with frequentist coverage that can be plugged into robust pipelines (Johnstone & Cox, 2021; Patel et al., 2024b). Our setting is complementary: rather than protecting an optimization model against uncertain parameters, we use conformal prediction sets as an interface for risk-averse decision making with action-conditional statistical guarantees; crucially, the conditioning event is endogenous (the selected action depends on the prediction set), so standard CRO calibration does not directly yield distribution-free action-conditional validity.

**Other Related Works.**   The max-min decision rule discussed in Section 2 naturally aligns with robust optimization paradigms, where optimal decisions are sought against worst-case realizations over uncertainty sets (Patel et al., 2024b; Johnstone & Cox, 2021; Chenreddy & Delage, 2024; Li et al., 2025; Yeh et al., 2024; Cao, 2024; Wang et al., 2025; Lou et al., 2024; Patel et al., 2024a; Elmachtoub et al., 2025; Lin et al., 2024; Chan et al., 2024; 2023; Chan & Kaw, 2020). However, these works typically do not consider action-conditional guarantees. Extending such robust formulations to incorporate action-conditioned safety constraints remains an intriguing direction for future work.

Conformal prediction's role in decision-making has also been examined from the perspective of predictive distributions (Vovk & Bendtsen, 2018; Vovk et al., 2017; 2018; 2020). These approaches generate calibrated distributions suitable for expectation-maximizing (i.e., risk-neutral) agents, and are more comparable to calibrated probabilistic forecasts. In contrast, we focus on risk-averse agents and show that prediction sets, rather than distributions, serve as surrogates for optimizing action-conditional value-at-risk objectives.

Finally, conformal prediction methods have been explored in a wide range of high-stakes applications. These include clinical medicine (Banerji et al., 2023), energy systems (Renkema et al., 2024), formal methods and control (Lindemann et al., 2024), and chance-constrained optimization (Zhao et al., 2024), UQ for LLMs (Quach et al., 2024; Noorani et al., 2026b; Kiyani et al., 2026), human-AI collaboration (Noorani et al., 2025; 2026a) among others (Sun et al., 2024; Ramalingam et al., 2024; Straitouri et al., 2023; Vishwakarma et al., 2024; van der Laan & Alaa, 2024; Noorani et al., 2024). We leave the investigation of application of our framework and action-conditional guarantees to these settings as future works.

## B. Details on the AC-RAC Algorithm

In the main text we outlined AC-RAC in Algorithm 1. Here we describe in detail how to solve the calibration subproblem

$$\hat{\boldsymbol{\lambda}}_y = \underset{\boldsymbol{\lambda} \in \mathbb{R}_{\geq 0}^{|\mathcal{A}|}}{\arg\min} F_y(\boldsymbol{\lambda}),$$

the core of which is minimizing the pinball-based loss $F_y$ defined in (15). Under a mild ties-breaking assumption, one can show that $F_y$ admits the following projected subgradient step for each action $a \in \mathcal{A}$:

$$\lambda_a \leftarrow \left[\lambda_a - \eta g_a\right]_+,$$

where $[\cdot]_+$ denotes projection onto the nonnegative reals, $\eta > 0$ is a step size, and

$$g_a = q_a\big(p_a - (1 - \alpha)\big), \quad q_a = \frac{n_a}{n+1}, \quad p_a = \frac{1}{n_a} \sum_{i:a_i=a} \mathbb{1}_{\{\lambda_a \geq \lambda_i^*\}}, \quad n_a = \big|\{i \leq n + 1 : a_i = a\}\big|.$$

Here:

- $a_i = \hat{a}\big(x_i, \hat{t}(x_i, \boldsymbol{\lambda})\big)$ is the action selected on calibration sample $(x_i, y_i)$ under the current thresholds.

- $\lambda_i^* = \lambda_{a_i}^*(x_i, y_i)$ is the action-conditional nonconformity score defined in Lemma 4.1. We remark that Lemma 4.1 implies that $p_a$ can be further rewritten as

$$p_a = \frac{\sum_{i:a_i=a} \mathbb{1}_{y_i \in \hat{C}\big(x_i, \boldsymbol{\lambda}\big)}}{n_a}.$$

- The test pair $(x_{\text{test}}, y)$ is treated as the $(n+1)$-th sample in all counts and sums.

Since each $g_a$ depends only on simple counts over the $n + 1$ points and the current actions $\{a_i\}$, it can be computed in $O(n|\mathcal{A}|)$ time per iteration. We present the computation of the full subgradient of $F_y$ in Lemma C.4.

## B.1. Pseudocode

---

**Algorithm 2** Solving AC-RAC with Gradient Descent

---

**Input:** Calibration samples $\{(x_i, y_i)\}_{i=1}^n$, test sample $(x_{n+1}, y_{n+1}) = (x_{\text{test}}, y)$, nominal coverage level $\alpha$, maximal number
      of iteration $K$, learning rates $\{\eta_k\}_{k=1}^K$

**for** $a \in \mathcal{A}$ **do**
    **Initialize dual:** $\lambda_a^{(0)} \leftarrow 0$

**for** $k \leftarrow 1$ **to** $K$ **do**
    **for** $i \leftarrow 1$ **to** $n + 1$ **do**
        **Select action:** $a_i \leftarrow \hat{a}\big(x_i, \hat{t}(x_i, \boldsymbol{\lambda}^{(k-1)})\big)$
        **Build set:** $C_i \leftarrow \hat{C}(x_i, \boldsymbol{\lambda}^{(k-1)})$
    **for** $a \in \mathcal{A}$ **do**
        **Estimate coverage:** $p_a \leftarrow \frac{|\{i : a_i = a \wedge y_i \in C_i\}|}{|\{i : a_i = a\}|}, q_a \leftarrow \frac{|\{i : a_i = a\}|}{n+1}$
        **Update parameter:** $\lambda_a^{(k)} \leftarrow \big[\lambda_a^{(k-1)} - \eta_k q_a(p_a - (1 - \alpha))\big]_+$

**return** $\hat{\boldsymbol{\lambda}}_y = \boldsymbol{\lambda}^{(K)}$

---

**Computational efficiency analysis.** In typical deep learning deployments, the dominant cost is computing $f_x$ for all calibration inputs once (a single forward pass per $x_i$), which is shared across all baselines. Given stored logits/probabilities, AC-RAC's additional overhead is the optimization over $\boldsymbol{\lambda}$. Algorithm 1 resembles label-conditional split conformal in that it performs an independent calibration for each candidate label $y \in \mathcal{Y}$, but our calibration involves solving a (sub)gradient-based optimization rather than a single quantile computation. Concretely, let $n$ be the calibration size, $K$ the number of subgradient steps used to minimize $F_y(\boldsymbol{\lambda})$, and $|\mathcal{A}|$ the number of actions. Each subgradient step recomputes the induced action $\hat{a}(x_i, \hat{t}(x_i, \boldsymbol{\lambda}))$ for $i \in [n]$ and evaluates the pinball losses, yielding total calibration time on the order of $\tilde{O}(|\mathcal{Y}| \cdot K \cdot n \cdot |\mathcal{A}|)$, where the $\tilde{O}(\cdot)$ notation hides the cost of the 1D maximizations over $t \in [0, 1]$ in (13) (and any fixed grid resolution used to evaluate $\hat{\theta}(x, t)$). This is higher than standard split conformal, but can be made practical via: (i) *precomputation*: the action-specific nonconformity scores $\lambda_a^*(x_i, y_i)$ depend only on $(x_i, y_i, a)$ and can be computed once and cached; (ii) *parallelization*: the calibrations across $y \in \mathcal{Y}$ are parallel; (iii) *warm starts*: initialize $\hat{\boldsymbol{\lambda}}_y$ from nearby labels or from a shared initialization; (iv) *stochastic subgradients*: use mini-batches of calibration points for large $n$.

## C. Technical Proofs

### C.1. Proof of Theorem 2.1

We proof the direction from AC-CPO to AC-DPO. Suppose $C(\cdot)$ is a feasible solution to Problem (7), i.e., for any $a \in \mathcal{A}$,

$$\mathbb{P}\big(Y \in C(X) \mid \arg\max_{a' \in \mathcal{A}} \min_{y \in C(X)} u(a', y) = a\big) \geq 1 - \alpha.$$

Define an action policy $a^*(x)$ and utility certificate $\nu^*(x)$ by

$$a^*(x) = \arg\max_{a \in \mathcal{A}} \min_{y \in C(x)} u(a, y), \qquad \nu^*(x) = \max_{a \in \mathcal{A}} \min_{y \in C(x)} u(a, y).$$

In the definition of $a^*(x)$, the maximization operator $\arg\max$ may return multiple actions if two or more actions attain the same value of the worst case utility. To ensure that $a^*(x)$ is a well-defined function, we impose a deterministic tie-breaking rule: among all maximizers, select one according to a fixed and consistent convention (for instance, by a pre-specified ordering of the action space or adding small gaussian noise to the utility function). This choice does not affect feasibility or the value of the objective, but it guarantees that the construction of $a^*(\cdot)$ and $\nu^*(\cdot)$ is unambiguous.

*Feasibility.* We claim that $(a^*(\cdot), \nu^*(\cdot))$ satisfies the AC-DPO constraints in (5). Note that $\nu^*(x)$ is exactly $\min_{y \in C(x)} u(a^*(x), y)$, by the definition of $a^*(x)$. Whenever $a^*(X) = a$, we have:

$$\{Y \in C(X)\} \implies \{u\big(a^*(X), Y\big) \geq \nu^*(X)\},$$

because if $Y \in C(X)$ then $u(a^*(X), Y) \geq \min_{y \in C(X)} u(a^*(X), y) = \nu^*(X)$. Hence,

$$\mathbb{P}\Big(u(a^*(X), Y) \geq \nu^*(X) \,\Big|\, a^*(X) = a\Big) \geq \mathbb{P}\Big(Y \in C(X) \,\Big|\, a^*(X) = a\Big) \geq 1 - \alpha,$$

where the last inequality is precisely the feasibility condition of $C(\cdot)$ in the AC-CPO problem. Therefore, $(a^*(\cdot), \nu^*(\cdot))$ is feasible for Problem (5).

*Objective Preservation.* Lastly, we compare the objective values of the two solutions. By definition,

$$\nu^*(x) = \max_{a \in \mathcal{A}} \min_{y \in C(x)} u(a, y).$$

But that is exactly the inner expression of the AC-CPO objective. Hence

$$\mathbb{E}_X\big[\nu^*(X)\big] = \mathbb{E}_X\Big[\max_{a \in \mathcal{A}} \min_{y \in C(X)} u(a, y)\Big],$$

which shows $(a^*(\cdot), \nu^*(\cdot))$ attains the same objective as $C(\cdot)$ in the AC-CPO problem. Thus, from any feasible solution $C(\cdot)$ of Problem (7), we can build a feasible solution $(a^*(\cdot), \nu^*(\cdot))$ of Problem (5) with the same objective value.

## C.2. Proofs for Section 3

### C.2.1. PROOF OF PROPOSITION 3.1

We prove the direction from (9) to (7). Let $t^*(x)$ be an optimal solution to (9). Define

$$a^*(x) = a(x, t^*(x)) = \arg\max_{a \in \mathcal{A}} \text{quantile}_{1-t^*(x)}[u(a, Y) \mid X = x],$$

$$\theta(x, t^*(x)) = \text{quantile}_{1-t^*(x)}[u(a^*(x), Y) \mid X = x],$$

and define the prediction set

$$C^*(x) = \{y \in \mathcal{Y} : u(a^*(x), y) \geq \theta(x, t^*(x))\}.$$

By construction of $\theta$ as a quantile, the set $C^*(x)$ satisfies

$$\mathbb{P}(Y \in C^*(x) \mid X = x) \geq t^*(x)$$

and

$$\min_{y \in C^*(x)} u(a^*(x), y) \geq \theta(x, t^*(x)).$$

*Remark* C.1 (Boundary ties and exact mass). When the conditional distribution of $u(a^*(x), Y) \mid X = x$ has atoms, the deterministic upper-quantile set $\{y : u(a^*(x), y) \geq \tau(x)\}$ may have probability mass strictly larger than $t^*(x)$. If one wishes to enforce exact conditional mass $t^*(x)$, a standard approach is randomized tie-breaking on the boundary: include points with $u(a^*(x), y) = \tau(x)$ with an independent Bernoulli coin flip (equivalently, add an arbitrarily small continuous jitter to break ties), choosing the tie-breaking probability so that $\mathbb{P}(Y \in C^*(x) \mid X = x) = t^*(x)$. Importantly, our feasibility and utility arguments do not rely on exact equality and remain valid using only $\mathbb{P}(Y \in C^*(x) \mid X = x) \geq t^*(x)$.

Next, we prove that for any $a \in \mathcal{A}$,

$$\text{quantile}_{1-t^*(x)}[u(a, Y) \mid X = x] \geq \min_{y \in C^*(x)} u(a, y).$$

We remark that the quantile function is defined as the upper quantile throught the paper, i.e.

$$\text{quantile}_{1-t^*(x)}[u(a, Y) \mid X = x] = \sup_z \{z : \mathbb{P}(u(a, Y) \geq z \mid X = x) \geq t^*(x)\}.$$

For any $a \in \mathcal{A}$, $\text{quantile}_{1-t^*(x)}[u(a, Y) \mid X = x] \geq \min_{y \in C^*(x)} u(a, y)$ is equivalent to

$$\mathbb{P}(u(a, Y) \geq \min_{y \in C^*(x)} u(a, y) \mid X = x) \geq t^*(x).$$

And the above inequality holds because $\{Y \in C^*(x)\} \subseteq \{u(a, Y) \geq \min_{y \in C^*(x)} u(a, y)\}$, hence

$$\mathbb{P}(u(a, Y) \geq \min_{y \in C^*(x)} u(a, y) \mid X = x) \geq \mathbb{P}(Y \in C^*(x) \mid X = x) \geq t^*(x).$$

Therefore, the above inequalities together show the equalities

$$\max_a \min_{y \in C^*(x)} u(a, y) = \theta(x, t^*(x))$$

and

$$\mathbb{E}_X \left[ \max_a \min_{y \in C^*(X)} u(a, y) \right] = \mathbb{E}_X[\theta(X, t^*(X))].$$

It remains to verify feasibility. Given the constraint in (9), we have

$$\mathbb{E}(t^*(X) \mid a(X, t^*(X)) = a) \geq 1 - \alpha, \quad \forall a.$$

If there exists $x$ such that $\arg\max_a \min_{y \in C^*(x)} u(a, y) \neq a^*(x)$, then we have

$$\max_a \min_{y \in C^*(x)} u(a, y) > \min_{y \in C^*(x)} u(a^*(x), y) = \theta(x, t^*(x))$$

and obtain the contradiction. As a result, the following events are equivalent

$$\{\arg\max_a \min_{y \in C^*(x)} u(a, y) = a\} = \{a(x, t^*(x)) = a\}$$

for all $a$ and $x$. Using $\mathbb{P}(Y \in C^*(x) \mid X = x) \geq t^*(x)$ and the tower rule, we obtain

$$\mathbb{P}(Y \in C^*(X) \mid \arg\max_a \min_{y \in C^*(X)} u(a, y) = a) = \mathbb{P}(Y \in C^*(X) \mid a^*(X) = a)$$

$$\geq \mathbb{E}(t^*(X) \mid a^*(X) = a) \geq 1 - \alpha, \quad \forall a. \tag{15}$$

### C.2.2. PROOF OF THEOREM 3.2

Our proof is comprised of three steps: linear reparametrization, continuous relaxation and strong duality. For the first step, we parametrize the objective function in (9) by the delta measure $\delta$ on $[0, 1]$; this technique allows us to rewrite the objective function and constraints in (9) into linear functionals with respect to $\delta$ (16). Next, since the Dirac measure is discrete and hard to handle, we relax this discrete optimization into the continuous regime (17) by expanding the domain of $\delta$ into all probability measures on $[0, 1]$. We prove that there is no relaxation gap. Finally, we consider the dual of the continuous optimization problem and prove strong duality.

**(I) Linear reparametrization.** We reparametrize (9) into a linear program by using a Dirac measure to rewrite $t(\cdot)$ in integral form. To begin with, define the Dirac measure $\delta(x, \cdot)$ on $[0, 1]$ for every $x \in \mathcal{X}$, where $\delta(x, \cdot)$ assigns a point mass at $t(x)$. In this case, we rewrite the function $t(\cdot)$ as:

$$t(x) = \int_0^1 s \, \delta(x, ds).$$

And the objective function in (9) can be rewritten as

$$\theta(x, t(x)) = \int_0^1 \theta(x, s) \, \delta(x, ds).$$

This technique allows us to rewrite the objective function and constraints in (9) into linear functionals with respect to $\delta$. In more detail, by the definition of conditional expectation, the coverage condition is equivalent to

$$\mathbb{E}_X \big[ t(X) \mathbb{1}_a(X, t(X)) \big] \geq (1 - \alpha) \mathbb{P}(a(X, t(X)) = a),$$

where $\mathbb{1}_a(X, t(X)) = 1$ iff $a(X, t(X)) = a$. Combining the above reparametrization, we arrive at the equivalent linear form for Problem (9):

$$\max_{\delta(x, \cdot): \text{ dirac measure on } [0,1] \text{ for all } x} \mathbb{E}_X \left[ \int_0^1 \theta(X, t) \, \delta(X, dt) \right],$$

$$\text{s.t.} \quad \mathbb{E}_X \left[ \int_0^1 t \, \mathbb{1}_a(X, t) \, \delta(X, dt) \right] \geq (1 - \alpha) \mathbb{E}_X \left[ \int_0^1 \mathbb{1}_a(X, t) \, \delta(X, dt) \right], \quad \forall a \in \mathcal{A}. \tag{16}$$

**(II) Continuous relaxation.** We remark that (16) is linear with respect to $\delta$ in terms of both objective function and constraints. Next, define $\Omega$ as the set of all measurable kernels $\delta : \mathcal{X} \to \mathcal{P}([0,1])$ such that, for each fixed $x$, $\delta(x, \cdot)$ is a probability measure on $[0,1]$. We relax the domain of $\delta$ to all probability measures to convert the problem into a continuous optimization:

$$\max_{\delta \in \Omega} \quad \mathbb{E}_X \left[ \int_0^1 \theta(X,t)\, \delta(X, dt) \right],$$

$$\text{s.t.} \quad \mathbb{E}_X \left[ \int_0^1 t\, \mathbb{1}_a(X,t)\, \delta(X, dt) \right] \geq (1-\alpha) \mathbb{E}_X \left[ \int_0^1 \mathbb{1}_a(X,t)\, \delta(X, dt) \right], \quad \forall a \in \mathcal{A}. \tag{17}$$

It is easy to see that this optimization problem has at least one feasible solution if we set $\delta(x, \cdot) = \delta_1$ (the point mass at $t = 1$), which corresponds to $C(x) = \mathcal{Y}$ for all $x$. Define

$$F(\delta) = \mathbb{E}_X \left[ \int_0^1 \theta(X,t)\, \delta(X, dt) \right]$$

and

$$G_a(\delta) = \mathbb{E}_X \left[ \int_0^1 t\, \mathbb{1}_a(X,t)\, \delta(X, dt) \right] - (1-\alpha) \mathbb{E}_X \left[ \int_0^1 \mathbb{1}_a(X,t)\, \delta(X, dt) \right]$$

for all $a \in \mathcal{A}$. Following standard Lagrangian duality for linear programs over measures, there exist Lagrange multipliers $\{\lambda_a\}_{a \in \mathcal{A}}$ with $\lambda_a \geq 0$ such that the dual objective can be written as

$$\Psi(\boldsymbol{\lambda}) = \sup_{\delta \in \Omega} \left\{ F(\delta) + \sum_a \lambda_a G_a(\delta) \right\}$$

$$= \sup_{\delta \in \Omega} \left\{ \mathbb{E}_X \int_0^1 \left( \theta(X,t) + \sum_a \lambda_a (t - (1-\alpha)) \mathbb{1}_a(X,t) \right) \delta(X, dt) \right\}. \tag{18}$$

Let

$$A(x,t) := \theta(x,t) + \sum_{a \in \mathcal{A}} \lambda_a \big( t - (1-\alpha) \big) \mathbb{1}_a(x,t).$$

Since $\delta(x, \cdot)$ can be chosen independently for each $x$ and $A(x,t)$ is bounded, the inner supremum over $\delta$ is attained by concentrating all mass on maximizers of $t \mapsto A(x,t)$, yielding

$$\sup_{\delta \in \Omega} \mathbb{E}_X \left[ \int_0^1 A(X,t)\, \delta(X, dt) \right] = \mathbb{E}_X \left[ \sup_{t \in [0,1]} A(X,t) \right].$$

For each fixed $x$, the inner problem reduces to

$$\sup_{\mu: \text{ prob. measure on } [0,1]} \int_0^1 A(x,t)\, \mu(dt). \tag{19}$$

**Lemma C.2.** *For any bounded upper-semicontinuous $A : [0,1] \to \mathbb{R}$, the optimal solution of*

$$\sup_{\mu: \text{ prob. measure on } [0,1]} \int_0^1 A(t)\, \mu(dt)$$

*is $\mu^* = \delta_{t^\star}$ for some $t^\star \in \arg\max_{s \in [0,1]} A(s)$, i.e. a point mass at a maximizer of $A$.*

*Proof.* Let $M = \sup_{t \in [0,1]} A(t) < \infty$. For any probability measure $\mu$ on $[0,1]$,

$$\int_0^1 A(t)\mu(dt) \leq \int_0^1 M\mu(dt) = M.$$

Since $A$ is upper-semicontinuous on the compact set $[0,1]$, there exists $t^\star \in \arg\max_{s \in [0,1]} A(s)$. Define $\mu^* = \delta_{t^\star}$. Then

$$\int_0^1 A(t)\mu^\star(dt) = A(t^\star) = M,$$

so $\mu^\star$ is optimal. $\qquad\square$

Using Lemma C.2, the optimizer of (19) can be taken as a Dirac measure at an optimizer of $t \mapsto A(x,t)$, i.e.,

$$\delta(x,\cdot) = \delta_{t^*(x,\boldsymbol{\lambda})}, \qquad t^*(x,\boldsymbol{\lambda}) \in \arg\max_{t \in [0,1]} \left( \theta(x,t) + \sum_a \lambda_a(t - (1-\alpha)) \mathbb{1}_a(x,t) \right). \tag{20}$$

Therefore, for any $\boldsymbol{\lambda} \geq 0$, the optimal $\delta^*(x,\cdot)$ and $t^*(x)$ can be computed by solving (20).

**(III) Strong duality.** The Lagrangian of (17) is $\mathcal{L}(\delta, \boldsymbol{\lambda}) = F(\delta) + \sum_a \lambda_a G_a(\delta)$. Taking the supremum over $\delta \in \Omega$ yields the dual objective

$$\Psi(\boldsymbol{\lambda}) = \sup_{\delta \in \Omega} \mathcal{L}(\delta, \boldsymbol{\lambda}) = \mathbb{E}_X\left[\varphi_X(\boldsymbol{\lambda})\right], \tag{21}$$

where

$$\varphi_X(\boldsymbol{\lambda}) = \max_{t \in [0,1]} \left( \theta(X,t) + \sum_{a \in \mathcal{A}} \lambda_a\left(t - (1-\alpha)\right) \mathbb{1}_a(X,t) \right).$$

We remark that the dual problem is

$$\min_{\boldsymbol{\lambda} \geq 0} \Psi(\boldsymbol{\lambda}). \tag{22}$$

Since (17) is a linear program over probability kernels and the feasible set is nonempty (e.g. $\delta(x,\cdot) = \delta_1$), and the objective is bounded, strong duality holds by standard linear-programming duality for measure-valued LPs. Hence the optimum of (17) equals the minimum of (22), and any dual minimizer $\boldsymbol{\lambda}^\star$ satisfies complementary slackness.

## C.3. Proofs for Section 4

### C.3.1. PROOF OF LEMMA 4.1

Fix a feature–label pair $(x,y)$ and a nonconformity score vector $\boldsymbol{\lambda} \in \mathbb{R}_{\geq 0}^{|\mathcal{A}|}$. Throughout the proof we suppress the symbol "^" on $\theta, a, t, C$ to lighten notation. To begin with, we need the following tie-breaking assumption.

**Assumption C.3.** Assume $\sum_{a \in \mathcal{A}} \mathbb{1}_{\{a(x,s)=a\}} = 1$ for every $(x,s) \in \mathcal{X} \times [0,1]$.

Intuitively, the assumption $\sum_{a \in \mathcal{A}} \mathbb{1}_{\{a(x,s)=a\}} = 1$ means that the quantile-based ranking of the utilities $\{\text{quantile}_{1-s}[u(a,Y) \mid Y \sim f_x]\}_{a \in \mathcal{A}}$ never ties. In degenerate cases one may add an arbitrarily small continuous perturbation to the utility before taking quantiles to break ties with probability 1.

**(I) Reduction to a single coordinate.** By Assumption C.3, $\sum_{a \in \mathcal{A}} \mathbb{1}_{\{a(x,t)=a\}} = 1$ for every $(x,t)$. Hence for each feature $x$ there exists a partition $\{S_a(x)\}_{a \in \mathcal{A}}$ of $[0,1]$ with

$$t \in S_a(x) \iff a(x,t) = a.$$

For $a \in \mathcal{A}$ consider the one-dimensional optimisation

$$T_a(x, \lambda_a) = \arg\max_{t \in S_a(x)} \left[\theta(x,t) + \lambda_a(t - (1-\alpha))\right].$$

We fix a deterministic tie–breaking rule and define the rightmost maximiser

$$t(x, \lambda_a) = \sup T_a(x, \lambda_a) \quad \text{(a single point)}.$$

Moreover, the function $H(t, \lambda_a) = \theta(x,t) + \lambda_a(t - (1-\alpha))$ has increasing differences in $(t, \lambda_a)$ because for $t' > t$,

$$H(t', \lambda_a) - H(t, \lambda_a) = \left(\theta(x,t') - \theta(x,t)\right) + \lambda_a(t' - t)$$

is nondecreasing in $\lambda_a$. Hence (by monotone comparative statics) the rightmost maximiser $\lambda_a \mapsto t(x, \lambda_a)$ is nondecreasing.

Since $\{S_a(x)\}_{a \in \mathcal{A}}$ partitions $[0,1]$, the global maximiser satisfies

$$t(x, \boldsymbol{\lambda}) \in \arg\max_{a \in \mathcal{A}} \max_{t \in S_a(x)} \left[\theta(x,t) + \lambda_a(t - (1-\alpha))\right],$$

and for each $a$ the inner maximiser is $t(x, \lambda_a)$ defined above. In particular, on the event $\{a(x, t(x, \boldsymbol{\lambda})) = a\}$ we have $t(x, \boldsymbol{\lambda}) = t(x, \lambda_a)$.

**(II) Label-dependent critical value.** Denote the one–sided upper quantile of the utility distribution induced by the predictive model $f_x$ by

$$q_{a,x}(t) = \sup\Big\{s \in \mathbb{R} : \mathbb{P}_{Y \sim f_x}\big(u(a, Y) \geq s\big) \geq t\Big\}, \qquad t \in [0, 1].$$

Then the quantile set at level $t$ is

$$\text{QuantileSet}_t[u(a, Y) \mid Y \sim f_x] = \{y \in \mathcal{Y} : u(a, y) \geq q_{a,x}(t)\}.$$

Recall the label-specific threshold

$$\lambda_a^*(x, y) = \inf\Big\{\lambda_a \geq 0 : y \in \text{QuantileSet}_{t(x,\lambda_a)}[u(a, Y) \mid Y \sim f_x]\Big\}.$$

Since $\lambda_a \mapsto t(x, \lambda_a)$ is nondecreasing and $q_{a,x}(\cdot)$ is nonincreasing, the mapping $\lambda_a \mapsto q_{a,x}\big(t(x, \lambda_a)\big)$ is nonincreasing. Hence the set inside the infimum is an interval of the form $[\lambda_a^*(x, y), \infty)$, and in particular

$$\lambda_a \geq \lambda_a^*(x, y) \iff u(a, y) \geq q_{a,x}\big(t(x, \lambda_a)\big) \iff y \in \text{QuantileSet}_{t(x,\lambda_a)}[u(a, Y) \mid Y \sim f_x]. \tag{23}$$

On the event $\{a(x, t(x, \boldsymbol{\lambda})) = a\}$, the conformal set equals

$$C(x, \boldsymbol{\lambda}) = \Big\{y \in \mathcal{Y} : u(a, y) \geq \theta\big(x, t(x, \boldsymbol{\lambda})\big)\Big\} = \Big\{y \in \mathcal{Y} : u(a, y) \geq q_{a,x}\big(t(x, \lambda_a)\big)\Big\},$$

where the last equality uses that on $S_a(x)$, by definition of $\theta$ and $a(x, t)$, $\theta(x, t) = q_{a,x}(t)$ and also $t(x, \boldsymbol{\lambda}) = t(x, \lambda_a)$ on the event. Combining this with (23) yields, conditional on $\{a(x, t(x, \boldsymbol{\lambda})) = a\}$,

$$\{y \in C(x, \boldsymbol{\lambda})\} \iff \{\lambda_a \geq \lambda_a^*(x, y)\}.$$

Replacing $y$ by the random label $Y$ gives the desired equivalence:

$$\{Y \in C(x, \boldsymbol{\lambda})\} = \{\lambda_a \geq \lambda_a^*(x, Y)\} \quad \text{conditional on} \quad \{a(x, t(x, \boldsymbol{\lambda})) = a\}.$$

### C.3.2. PROOF OF THEOREM 4.3

Throughout the proof we suppress the symbol "^" on $\theta, a, t, C$ to lighten notation. We denote $(x_{n+1}, y_{n+1}) = (x_{\text{test}}, y_{\text{test}})$ for simplicity.

For any fixed $\boldsymbol{\lambda}$ and index $i \in [n + 1]$, define $a_i(\boldsymbol{\lambda}) = a(x_i, t(x_i, \boldsymbol{\lambda}))$, define the weighted sum of indicator functions

$$z_i(\boldsymbol{\lambda}) = \sum_{a \in \mathcal{A}} \lambda_a \mathbb{1}_{\{a(x_i, t(x_i, \boldsymbol{\lambda})) = a\}}, \qquad z_i^*(\boldsymbol{\lambda}) = \sum_{a \in \mathcal{A}} \lambda_a^*(x_i, y_i) \mathbb{1}_{\{a(x_i, t(x_i, \boldsymbol{\lambda})) = a\}},$$

and denote

$$\mathcal{B}_a(\boldsymbol{\lambda}) = \big\{i \leq n + 1 : a_i = a, z_i(\boldsymbol{\lambda}) = z_i^*(\boldsymbol{\lambda})\big\}, \qquad B_a(\boldsymbol{\lambda}) = |\mathcal{B}_a(\boldsymbol{\lambda})|.$$

**Lemma C.4.** *Under the tie breaking assumption C.3, for any $y = y_{n+1} \in \mathcal{Y}$, the sub-gradient of the empirical risk $F_y(\boldsymbol{\lambda})$:*

$$F_y(\boldsymbol{\lambda}) = \frac{1}{n+1} \sum_{i=1}^{n+1} \ell_\alpha\big(z_i(\boldsymbol{\lambda}), z_i^*(\boldsymbol{\lambda})\big), \qquad \ell_\alpha(u, v) = (1 - \alpha)(v - u)_+ + \alpha(u - v)_+,$$

*is the set of vectors (the axis-parallel interval product)*

$$\partial F_y(\boldsymbol{\lambda}) = \Big(q_a\big(p_a - (1 - \alpha)\big) + [-B_a(\boldsymbol{\lambda})/(n + 1), 0]\Big)_{a \in \mathcal{A}}$$

*where*

$$q_a = \frac{n_a}{n+1}, \qquad p_a = \frac{\sum_{i:a_i=a} \mathbb{1}_{\{z_i \geq z_i^*\}}}{n_a}, \qquad n_a = |\{i \leq n + 1 : a_i = a\}|.$$

*Proof.* For each calibration index $i \leq n+1$ write

$$r_i(\boldsymbol{\lambda}) = z_i(\boldsymbol{\lambda}) - z_i^*(\boldsymbol{\lambda}) = \sum_{a \in \mathcal{A}} \big(\lambda_a - \lambda_a^*(x_i, y_i)\big) \mathbb{1}_{\{a_i(\boldsymbol{\lambda})=a\}}, \quad a_i(\boldsymbol{\lambda}) = a\big(x_i, t(x_i, \boldsymbol{\lambda})\big).$$

Because $F_y$ is a sum of composite functions $\ell_\alpha(r_i, 0)$, the Clarke chain rule gives, for every coordinate $a \in \mathcal{A}$,

$$\partial_{\lambda_a} F_y(\boldsymbol{\lambda}) = \frac{1}{n+1} \sum_{i=1}^{n+1} s_i(\boldsymbol{\lambda}) \, g_{i,a}(\boldsymbol{\lambda}), \tag{24}$$

with

$$s_i(\boldsymbol{\lambda}) \in \partial_u \ell_\alpha(u, 0)\big|_{u=r_i(\boldsymbol{\lambda})}, \quad g_{i,a}(\boldsymbol{\lambda}) \in \partial_{\lambda_a} r_i(\boldsymbol{\lambda}).$$

We remark that the pin-ball loss has one–dimensional sub-differential

$$s_i(\boldsymbol{\lambda}) \in \begin{cases} \{\alpha\}, & r_i(\boldsymbol{\lambda}) > 0, \\ [\alpha - 1, \, \alpha], & r_i(\boldsymbol{\lambda}) = 0, \\ \{-(1-\alpha)\}, & r_i(\boldsymbol{\lambda}) < 0. \end{cases}$$

Now we calculate $g_{i,a}(\boldsymbol{\lambda})$. Because of the tie–breaking rule, the maximising action $a_i(\boldsymbol{\lambda})$ is locally constant away from switching points, and hence $r_i$ is locally affine there. At switching points, $g_{i,a}(\boldsymbol{\lambda})$ is understood in the Clarke sense, i.e., it lies in the convex hull of neighboring gradients; this does not affect the interval form derived below. Thus we may take

$$g_{i,a}(\boldsymbol{\lambda}) \in \partial_{\lambda_a} r_i(\boldsymbol{\lambda}) = \begin{cases} 1, & a = a_i(\boldsymbol{\lambda}), \\ 0, & a \neq a_i(\boldsymbol{\lambda}). \end{cases} \tag{25}$$

Since $g_{i,a}$ equals 1 if $a_i = a$ and 0 otherwise, the chain rule (24) simplifies to

$$\partial_{\lambda_a} F_y(\boldsymbol{\lambda}) = \frac{1}{n+1} \sum_{i:a_i=a} s_i(\boldsymbol{\lambda}).$$

Partition the indices with action $a$ as

$$\mathcal{I}_a^+ = \{i : r_i > 0\}, \ \mathcal{I}_a^- = \{i : r_i < 0\}, \ \mathcal{B}_a = \{i : r_i = 0\}, \quad n_a = |\mathcal{I}_a^+| + |\mathcal{I}_a^-| + B_a.$$

On $\mathcal{I}_a^+$ we have $s_i = \alpha$, on $\mathcal{I}_a^-$ we have $s_i = -(1-\alpha)$, and on each boundary index $i \in \mathcal{B}_a$ we may choose an arbitrary value $s_i = -(1-\alpha) + h_i$, $h_i \in [0,1]$. Therefore

$$\partial_{\lambda_a} F_y(\boldsymbol{\lambda}) = \frac{1}{n+1} \Big(\alpha|\mathcal{I}_a^+| - (1-\alpha)|\mathcal{I}_a^-| - (1-\alpha)B_a + \sum_{i \in \mathcal{B}_a} h_i\Big).$$

By the definition, we have

$$q_a = \frac{n_a}{n+1}, \qquad p_a = \frac{|\mathcal{I}_a^+|}{n_a},$$

so that $\alpha|\mathcal{I}_a^+| - (1-\alpha)|\mathcal{I}_a^-| = (n_a)\big(p_a - (1-\alpha)\big)$. The "anchor" part of the sub-gradient is therefore $q_a\big(p_a - (1-\alpha)\big)$. Finally, because $\sum_{i \in \mathcal{B}_a} h_i \in [0, B_a]$, the boundary indices create the extra interval $\big[-B_a/(n+1), 0\big]$, which yields the claimed product form. $\square$

Now back to the proof of Theorem 4.3. Suppose $\boldsymbol{\lambda}$ is the minimizer of $F_{y_{n+1}}(\boldsymbol{\lambda})$, then we have

$$0 \in \partial F_{y_{n+1}}(\boldsymbol{\lambda}).$$

Using Lemma C.4, we have arrived at

$$0 \in \partial_{\lambda_a} F_{y_{n+1}}(\boldsymbol{\lambda}) = q_a(p_a - (1-\alpha)) + [-B_a(\boldsymbol{\lambda})/(n+1), 0], \quad \forall a \in \mathcal{A}. \tag{26}$$

Besides, we rewrite the coverage $\mathbb{P}(y_{n+1} \in C(x_{n+1}, \boldsymbol{\lambda}), a_{n+1}(\boldsymbol{\lambda}) = a)$ as follows:

$$\mathbb{P}(y_{n+1} \in C(x_{n+1}, \boldsymbol{\lambda}), a_{n+1}(\boldsymbol{\lambda}) = a)$$

$$= \mathbb{E}\big[\mathbb{1}_{Y_{n+1} \in C(x_{n+1}, \boldsymbol{\lambda})} \mathbb{1}_{a_{n+1}(\boldsymbol{\lambda}) = a}\big]$$

$$= \frac{1}{n+1} \cdot \mathbb{E}\left[\sum_{i=1}^{n+1} \mathbb{1}_{y_i \in C(x_i, \boldsymbol{\lambda})} \mathbb{1}_{a_i(\boldsymbol{\lambda}) = a}\right].$$

The last equation holds because data are exchangable. Recall the definition of $p_a$ and $q_a$, Lemma 4.1 implies that

$$\frac{1}{n+1} \cdot \sum_{i=1}^{n+1} \mathbb{1}_{y_i \in C(x_i, \boldsymbol{\lambda})} \mathbb{1}_{a_i(\boldsymbol{\lambda}) = a} = p_a \cdot q_a.$$

Therefore, plugging the closed-form of $\partial_{\lambda_a} F_{y_{n+1}}(\boldsymbol{\lambda})$ in (26), we obtain that

$$0 \leq q_a(p_a - (1 - \alpha)) \leq \frac{\sum_a B_a(\boldsymbol{\lambda})}{n+1}.$$

Using exchangeability and the identity $\mathbb{P}(y_{n+1} \in C(x_{n+1}, \boldsymbol{\lambda}), a_{n+1}(\boldsymbol{\lambda}) = a) = \mathbb{E}[q_a p_a]$ and $\mathbb{P}(a_{n+1}(\boldsymbol{\lambda}) = a) = \mathbb{E}[q_a]$, we conclude

$$0 \leq \mathbb{P}(y_{n+1} \in C(x_{n+1}, \boldsymbol{\lambda}), a_{n+1}(\boldsymbol{\lambda}) = a) - (1 - \alpha) \cdot \mathbb{P}(a_{n+1}(\boldsymbol{\lambda}) = a) = \mathbb{E}[q_a(p_a - (1 - \alpha))] \leq \frac{\sum_a B_a(\boldsymbol{\lambda})}{n+1}.$$

We note that if $(x_i, y_i)$ are i.i.d. and the nonconformity score $\lambda_a^*(x_i, y_i)$ has a continuous distribution, the tie happens with probability zero and $|B_a(\boldsymbol{\lambda})| = 1$ for all $a$ with probability 1. Hence

$$0 \leq \mathbb{E}[q_a(p_a - (1 - \alpha))] \leq \frac{|\mathcal{A}|}{n+1}.$$

Dividing by $\mathbb{P}(a_{n+1}(\boldsymbol{\lambda}) = a)$ (assuming it is positive) yields

$$1 - \alpha \leq \mathbb{P}\big(Y_{n+1} \in C_{n+1} \mid a_{n+1} = a\big) \leq 1 - \alpha + \frac{|\mathcal{A}|}{(n+1) \cdot \mathbb{P}(a_{n+1} = a)}.$$

### C.4. Convergence of Algorithm 2

In this section, we prove the convergence of the gradient steps in Algorithm 2.

**Theorem C.5.** *Let $\{\boldsymbol{\lambda}^{(k)}\}_{k \geq 0}$ be produced by the update rule in Algorithm 2 with stepsizes $\{\eta_k\}_{k \geq 0}$ obeying*

$$\eta_k > 0, \qquad \sum_{k=0}^{\infty} \eta_k = \infty, \qquad \sum_{k=0}^{\infty} \eta_k^2 < \infty. \tag{27}$$

*Then for any $y \in \mathcal{Y}$, Algorithm 2 has the following convergence guarantees:*

$$\liminf_{k \to \infty} F_y(\boldsymbol{\lambda}^{(k)}) \to F_y^\star, \quad \text{where} \quad F_y^\star = \min_{\boldsymbol{\lambda} \geq 0} F_y(\boldsymbol{\lambda}).$$

*Proof.* For each coordinate $a$ and every $k$, Lemma C.4 gives a valid sub-gradient

$$g_a^{(k)} = \partial_{\lambda_a} F_y\big(\boldsymbol{\lambda}^{(k)}\big) = q_a^{(k)}\big(p_a^{(k)} - (1 - \alpha)\big),$$

where $p_a^{(k)}, q_a^{(k)}$ are induced by $\boldsymbol{\lambda}^{(k)}$. We note that $0 \leq p_a^{(k)}, q_a^{(k)} \leq 1$, so $\|g^{(k)}\|_2 \leq G = \sqrt{|\mathcal{A}|}$ for all $k$.

Let $P_+(\cdot)$ be the projection onto the non-negative orthant $\Lambda = \mathbb{R}_{\geq 0}^{|\mathcal{A}|}$ (non-expansive). Writing $\boldsymbol{\lambda}^{(k+1)} = P_+\big(\boldsymbol{\lambda}^{(k)} - \eta_k g^{(k)}\big)$, for any $\boldsymbol{\lambda} \in \Lambda$

$$\big\|\boldsymbol{\lambda}^{(k+1)} - \boldsymbol{\lambda}\big\|_2^2 \leq \big\|\boldsymbol{\lambda}^{(k)} - \eta_k g^{(k)} - \boldsymbol{\lambda}\big\|_2^2 = \big\|\boldsymbol{\lambda}^{(k)} - \boldsymbol{\lambda}\big\|_2^2 - 2\eta_k \langle g^{(k)}, \boldsymbol{\lambda}^{(k)} - \boldsymbol{\lambda}\rangle + \eta_k^2 G^2.$$

*Table 1.* Utility matrix for medical diagnosis actions.

| True Label | No Action | Antibiotics | Quarantine | Additional Testing |
|---|---|---|---|---|
| Normal (0) | 10 | 2 | 2 | 4 |
| Pneumonia (1) | 0 | 10 | 3 | 7 |
| COVID-19 (2) | 0 | 3 | 10 | 8 |
| Lung Opacity (3) | 1 | 4 | 4 | 10 |

*Table 2.* Utility matrix for the MovieLens recommendation task.

| Action | Rating 1 | Rating 2 | Rating 3 | Rating 4 | Rating 5 |
|---|---|---|---|---|---|
| No-Recommend | 0 | 0 | 0 | 0 | 0 |
| Recommend | -2 | -1 | 0 | +1 | +2 |

Choose $\boldsymbol{\lambda} = \boldsymbol{\lambda}^\star \in \arg\min F_y$ (which exists because $F_y \geq 0$ and $\Lambda$ is closed and coercive in $\boldsymbol{\lambda}$). Because $g^{(k)}$ is a sub-gradient,

$$F_y(\boldsymbol{\lambda}^{(k)}) - F_y(\boldsymbol{\lambda}^\star) \leq \langle g^{(k)}, \boldsymbol{\lambda}^{(k)} - \boldsymbol{\lambda}^\star \rangle.$$

Insert this bound and sum the resulting inequality from $k = 0$ to $K - 1$:

$$2 \sum_{k=0}^{K-1} \eta_k \big( F_y(\boldsymbol{\lambda}^{(k)}) - F_y^\star \big) \leq \big\| \boldsymbol{\lambda}^{(0)} - \boldsymbol{\lambda}^\star \big\|_2^2 + G^2 \sum_{k=0}^{K-1} \eta_k^2.$$

The right–hand side is finite by $\sum_k \eta_k^2 < \infty$, while the left–hand side is a sum of non–negative terms. Hence $\sum_{k=0}^{\infty} \eta_k \big( F_y(\boldsymbol{\lambda}^{(k)}) - F_y^\star \big) < \infty$. Because $\sum_k \eta_k = \infty$, the only way the weighted sum can be finite is if

$$\liminf_{k \to \infty} F_y(\boldsymbol{\lambda}^{(k)}) \to F_y^\star.$$

$\square$

# D. Details on Numerical Experiments

## D.1. Experiment details

In this section, we present the details of numerical experiments. All the experiments are conduct on a computation cluster using a RTX 5080 and AMD 9950X3D. In the medical diagnosis experiment, the maximal number of iteration is 300 and the learning rate $\eta_k$ is 5. In the recommender system experiment, the maximal number of iteration is 200 and the learning rate $\eta_k$ is 30. We set the nominal level as 0.05 and run both of the experiments over 40 random seeds to plot Figure 1.

## D.2. Utility Functions

We present Table 1 for the utility matrix of the medical diagnosis experiment and Table 2 for the utility matrix of the recommender systems experiment. We remark that the utility functions are adopt from (Kiyani et al., 2025b), where the utility matrix are generated by Chatgpt and can be replaced to any other user-specified matrices. Besides, we add a simple tie breaking procedure in the recommender systems experiment. In more detail, we use $\epsilon \cdot (3 - R)$ as the true utility function, where $R$ denotes the rating and $\epsilon = 0.1$ is a small perturbation value.

## D.3. Additional Empirical Diagnostics

This section collects the empirical diagnostics that address three practical questions: whether action-conditioning substantially enlarges prediction sets, whether the missing action in Figure 1 reflects undefined conditional errors for baselines, and whether AC-RAC remains stable under rare actions and moderately larger finite action spaces. Unless otherwise noted, all diagnostics use the COVID medical-diagnosis setup.

*Table 3.* Mean prediction-set size on the COVID task across nominal levels.

| $\alpha$ | AC-RAC | RAC | score-1 | score-2 |
|------|--------|-------|---------|---------|
| 0.01 | 3.797 | 3.773 | 2.980 | 2.957 |
| 0.02 | 3.701 | 3.646 | 2.802 | 2.812 |
| 0.03 | 3.566 | 3.487 | 2.664 | 2.681 |
| 0.05 | 3.373 | 3.252 | 2.468 | 2.419 |
| 0.10 | 2.964 | 2.750 | 2.102 | 2.058 |

*Table 4.* False discovery rate (FDR) and mean prediction-set size at $\alpha = 0.05$ on the COVID task.

| Method | Mean prediction-set size | Overall FDR |
|--------|--------------------------|-------------|
| AC-RAC | 3.373 | 0.699 |
| RAC | 3.252 | 0.682 |
| score-1 | 2.468 | 0.585 |
| score-2 | 2.419 | 0.550 |

Tables 3 and 4 show that AC-RAC is not obtaining stronger action-conditional guarantees through a dramatic increase in conservativeness. At $\alpha = 0.05$, the mean set size increases from 3.252 for RAC to 3.373 for AC-RAC, a relative increase of about 3.7%, while the overall FDR changes only modestly from 0.682 to 0.699. The smaller FDR values of the action-independent score baselines should be interpreted together with action usage: these methods often avoid difficult actions rather than calibrating them reliably.

*Table 5.* Test-time action frequencies on the COVID task at $\alpha = 0.05$.

| Method | No Action | Antibiotics | Quarantine | Additional Testing |
|--------|-----------|-------------|------------|--------------------|
| AC-RAC | 4.87% | 1.11% | 2.72% | 91.31% |
| RAC | 30.94% | 0.68% | 0.00% | 68.38% |
| score-1 | 4.72% | 0.14% | 0.00% | 95.13% |
| score-2 | 12.45% | 0.52% | 0.00% | 87.03% |
| Best Response | 44.45% | 1.82% | 0.00% | 53.73% |

Table 5 explains why action 2 is absent for the baselines in the medical-diagnosis miscoverage panel. AC-RAC is the only method that selects Quarantine, doing so on 115 out of 4234 test instances. This illustrates the phenomenon AC-RAC is designed to address: marginally calibrated or risk-neutral procedures can appear acceptable on average while collapsing onto dominant actions and leaving rare or difficult actions without meaningful conditional guarantees.

*Table 6.* Rare-action stress test on the COVID task.

| Target prevalence | Method | Target-action coverage | Overall coverage | Mean set size |
|-------------------|--------|------------------------|------------------|---------------|
| 5.3% (full) | AC-RAC | 0.927 | 0.946 | 3.373 |
| 5.3% (full) | RAC | 0.871 | 0.953 | 3.252 |
| 1.0% | AC-RAC | 0.934 | 0.945 | 3.350 |
| 1.0% | RAC | 0.871 | 0.952 | 3.242 |
| 0.5% | AC-RAC | 0.931 | 0.945 | 3.341 |
| 0.5% | RAC | 0.871 | 0.952 | 3.242 |
| 0.1% | AC-RAC | 0.934 | 0.945 | 3.349 |
| 0.1% | RAC | 0.871 | 0.952 | 3.242 |

Table 6 shows that AC-RAC does not collapse when the target action becomes rare in calibration. Its target-action coverage

remains around 0.93–0.94 and its mean prediction-set size stays essentially unchanged. We present this as a controlled stress test, since the predictive scores are fixed and only the calibration prevalence is perturbed.

*Table 7.* Controlled scaling ablation on the COVID task with enlarged finite action spaces.

| $|\mathcal{A}|$ | Method | Mean set size | Overall coverage | Critical error rate | Actions used |
|---|---|---|---|---|---|
| 4 | AC-RAC | 3.373 | 0.946 | 0.35% | 4 |
| 4 | RAC | 3.252 | 0.953 | 3.99% | 3 |
| 6 | AC-RAC | 3.042 | 0.968 | 0.21% | 6 |
| 6 | RAC | 3.128 | 0.957 | 3.35% | 5 |
| 8 | AC-RAC | 3.643 | 0.981 | 0.21% | 8 |
| 8 | RAC | 3.136 | 0.951 | 4.02% | 6 |
| 10 | AC-RAC | 3.415 | 0.967 | 0.21% | 9 |
| 10 | RAC | 3.187 | 0.950 | 4.70% | 9 |

Table 7 suggests that the main empirical picture remains stable as the finite action space grows in this controlled regime. The mean prediction-set size remains in a narrow range rather than exploding with $|\mathcal{A}|$, AC-RAC continues to use nearly all available actions, and the critical error rate remains very low (0.21–0.35%). By contrast, RAC has substantially higher critical error rates (3.35–4.70%) and uses fewer actions in the smaller action spaces. This experiment is not a claim that large, continuous, or combinatorial action spaces are solved; rather, it probes moderate finite-action scalability under a hand-crafted enlarged utility matrix.

### D.4. Supplementary Figures

Figure 2 provides a detailed report of the medical diagnosis experiment:

- **Top row (a)–(d):** Test-time miscoverage curves for four decision rules (AC-RAC in red, RAC in blue, score-1 in orange, score-2 in green) plotted against the nominal miscoverage level $\alpha$ for each action $a \in \{0, 1, 2, 3\}$. The dashed grey line shows the identity miscoverage $= \alpha$.

- **Bottom left (e):** Average realized utility as a function of $\alpha$ for the same four methods together with the best-response baseline (black). All panels share a consistent color and marker scheme.

Figure 2 implies that AC-RAC's curves in panels (a)–(d) are close to the identity line across all actions and all nominal levels, whereas the baselines systematically deviate—over-covering some actions and under-covering others. This verifies that only AC-RAC enforces the desired conditional coverage guarantee in finite samples.

Figure 3 shows the corresponding results for the recommender system experiment:

- **Top row (a)–(b):** Test-time miscoverage for actions No-Rec and Rec.

- **Bottom (c):** Average realized utility vs. nominal $\alpha$.

Figure 3 implies that AC-RAC is the only method whose miscoverage curves lie at or below the nominal line for both actions, confirming reliable action-conditional coverage. In panel (c), AC-RAC's average utility increases smoothly as $\alpha$ shrinks, remaining within 5% of RAC (which is marginally optimal) and significantly outperforming score-1 and score-2. This demonstrates that AC-RAC preserves high decision quality despite its stricter safety requirements.

Taken together, these supplementary figures reinforce the main text's conclusion: AC-RAC uniquely achieves action-conditional coverage while preserving robust utility across diverse settings.

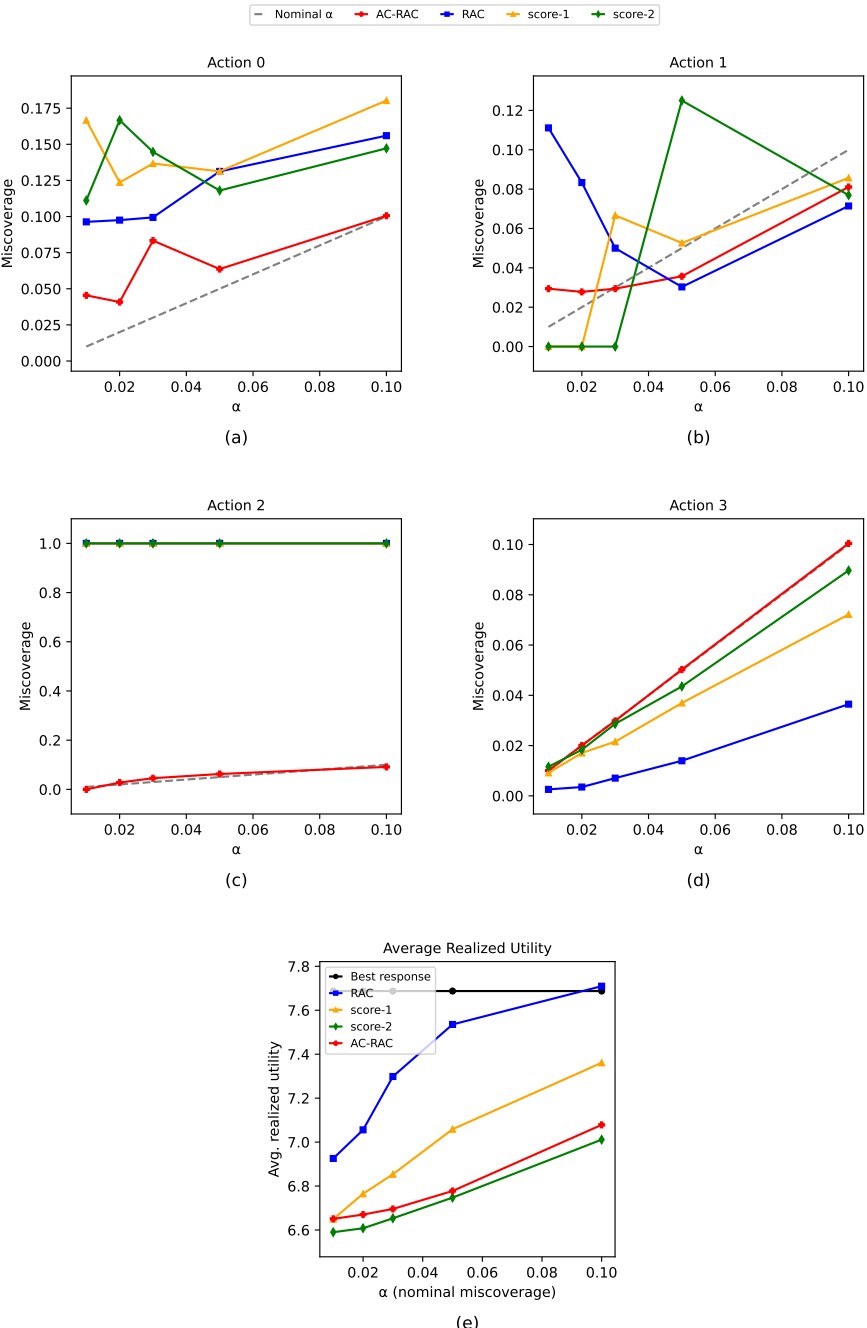

*Figure 2.* (a–d) Miscoverage vs. $\alpha$ for each action under AC-RAC (red), RAC (blue), s1 (orange), s2 (green) (dashed identity). (e) Average utility vs. $\alpha$.

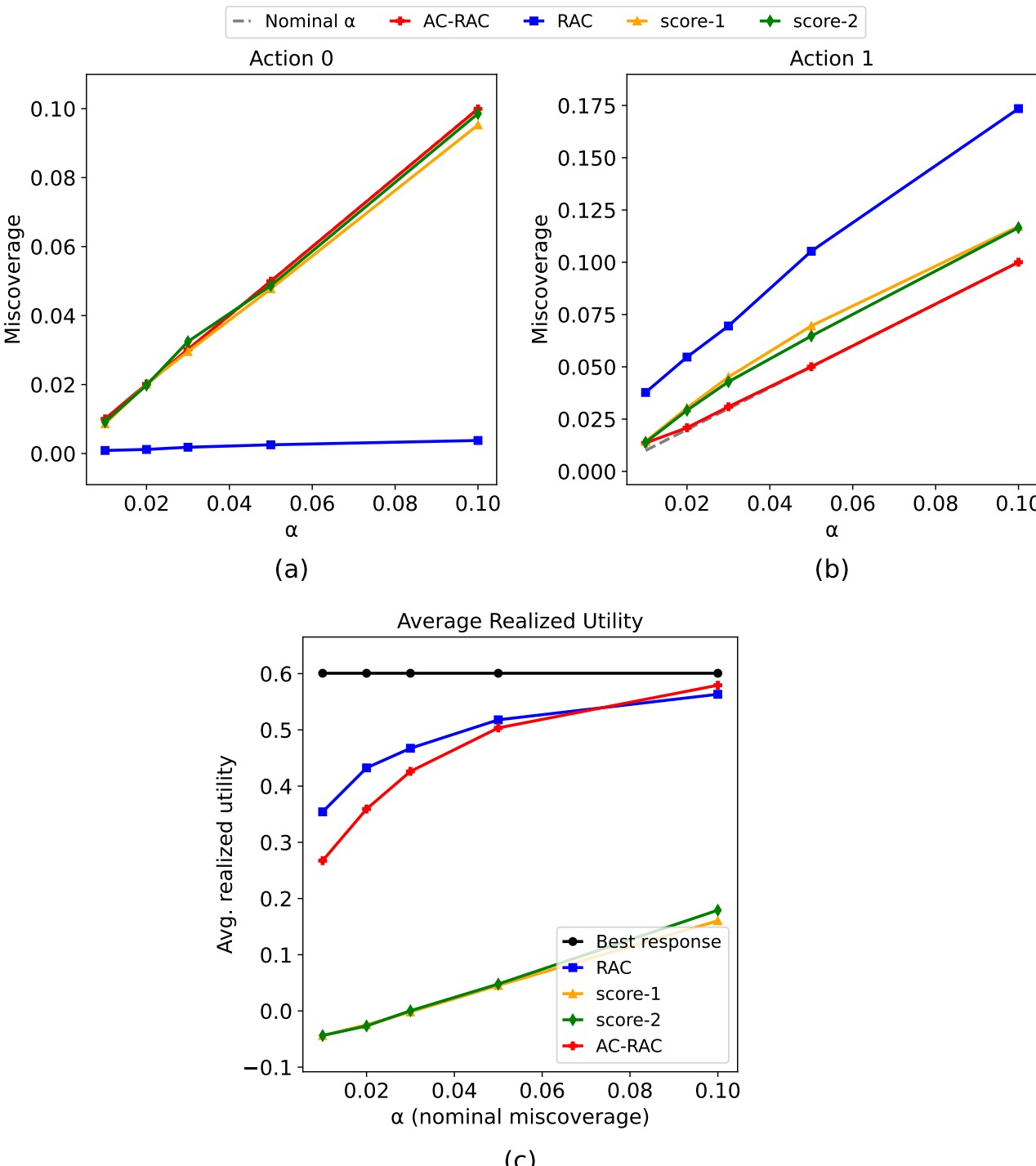

*Figure 3.* (a–b) Miscoverage vs. $\alpha$ for each action under AC-RAC (red), RAC (blue), s1 (orange), s2 (green) (dashed identity). (c) Average utility vs. $\alpha$.

