# Conformal Risk-Averse Decision Making with Action Conditional Guarantee

## Abstract

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

_+(\boldsymbol{\lambda}^{(k)} - \eta_k g^{(k)})$, for any $\boldsymbol{\lambda} \in \Lambda$

$$\|\boldsymbol{\lambda}^{(k+1)} - \boldsymbol{\lambda}\|_2^2 \leq \|\boldsymbol{\lambda}^{(k)} - \eta_k g^{(k)} - \boldsymbol{\lambda}\|_2^2 = \|\boldsymbol{\lambda}^{(k)} - \boldsymbol{\lambda}\|_2^2 - 2\eta_k \langle g^{(k)}, \boldsymbol{\lambda}^{(k)} - \boldsymbol{\lambda}\rangle + \eta_k^2 G^2.$$

*Table 1.* Utility matrix for medical diagnosis actions.

| True Label | No Action | Antibiotics | Quarantine | Additional Testing |
|---|---|---|---|---|
| Normal (0) | 10 | 2 | 2 | 4 |
| Pneumonia (1) | 0 | 10 | 3 | 7 |
| COVID-19 (2) | 0 | 3 | 10 | 8 |
| Lung Opacity (3) | 1 | 4 | 4 | 10 |

*Table 2.* Utility matrix for the MovieLens recommendation task.

| Action | Rating 1 | Rating 2 | Rating 3 | Rating 4 | Rating 5 |
|---|---|---|---|---|---|
| No-Recommend | 0 | 0 | 0 | 0 | 0 |
| Recommend | -2 | -1 | 0 | +1 | +2 |

Choose $\boldsymbol{\lambda} = \boldsymbol{\lambda}^\star \in \arg\min F_y$ (which exists because $F_y \geq 0$ and $\Lambda$ is closed and coercive in $\boldsymbol{\lambda}$). Because $g^{(k)}$ is a sub-gradient,

$$F_y(\boldsymbol{\lambda}^{(k)}) - F_y(\boldsymbol{\lambda}^\star) \leq \langle g^{(k)}, \boldsymbol{\lambda}^{(k)} - \boldsymbol{\lambda}^\star \rangle.$$

Insert this bound and sum the resulting inequality from $k = 0$ to $K - 1$:

$$2 \sum_{k=0}^{K-1} \eta_k \big( F_y(\boldsymbol{\lambda}^{(k)}) - F_y^\star \big) \leq \big\| \boldsymbol{\lambda}^{(0)} - \boldsymbol{\lambda}^\star \big\|_2^2 + G^2 \sum_{k=0}^{K-1} \eta_k^2.$$

The right–hand side is finite by $\sum_k \eta_k^2 < \infty$, while the left–hand side is a sum of non–negative terms. Hence $\sum_{k=0}^\infty \eta_k \big( F_y(\boldsymbol{\lambda}^{(k)}) - F_y^\star \big) < \infty$. Because $\sum_k \eta_k = \infty$, the only way the weighted sum can be finite is if

$$\liminf_{k \to \infty} F_y(\boldsymbol{\lambda}^{(k)}) \to F_y^\star.$$

$\square$

# D. Details on Numerical Experiments

## D.1. Experiment details

In this section, we present the details of numerical experiments. All the experiments are conduct on a computation cluster using a RTX 5080 and AMD 9950X3D. In the medical diagnosis experiment, the maximal number of iteration is 300 and the learning rate $\eta_k$ is 5. In the recommender system experiment, the maximal number of iteration is 200 and the learning rate $\eta_k$ is 30. We set the nominal level as 0.05 and run both of the experiments over 50 random seeds to plot Figure 1.

## D.2. Utility Functions

We present Table 1 for the utility matrix of the medical diagnosis experiment and Table 2 for the utility matrix of the recommender systems experiment. We remark that the utility functions are adopt from (Kiyani et al., 2025), where the utility matrix are generated by Chatgpt and can be replaced to any other user-specified matrices. Besides, we add a simple tie breaking procedure in the recommender systems experiment. In more detail, we use $\epsilon \cdot (3 - R)$ as the true utility function, where $R$ denotes the rating and $\epsilon = 0.1$ is a small perturbation value.

## D.3. Supplementary Figures

Figure 2 provides a detailed report of the medical diagnosis experiment:

- **Top row (a)–(d):** Test-time miscoverage curves for four decision rules (AC-RAC in red, RAC in blue, score-1 in orange, score-2 in green) plotted against the nominal miscoverage level $\alpha$ for each action $a \in \{0, 1, 2, 3\}$. The dashed grey line shows the identity miscoverage $= \alpha$.

- **Bottom left (e):** Average realized utility as a function of $\alpha$ for the same four methods together with the best-response baseline (black). All panels share a consistent color and marker scheme.

Figure 2 implies that AC-RAC's curves in panels (a)–(d) are close to the identity line across all actions and all nominal levels, whereas the baselines systematically deviate—over-covering some actions and under-covering others. This verifies that only AC-RAC enforces the desired conditional coverage guarantee in finite samples.

Figure 3 shows the corresponding results for the recommender system experiment:

- **Top row (a)–(b):** Test-time miscoverage for actions No-Rec and Rec.

- **Bottom (c):** Average realized utility vs. nominal $\alpha$.

Figure 3 implies that AC-RAC is the only method whose miscoverage curves lie at or below the nominal line for both actions, confirming reliable action-conditional coverage. In panel (c), AC-RAC's average utility increases smoothly as $\alpha$ shrinks, remaining within 5% of RAC (which is marginally optimal) and significantly outperforming score-1 and score-2. This demonstrates that AC-RAC preserves high decision quality despite its stricter safety requirements.

Taken together, these supplementary figures reinforce the main text's conclusion: AC-RAC uniquely achieves action-conditional coverage while preserving robust utility across diverse settings.

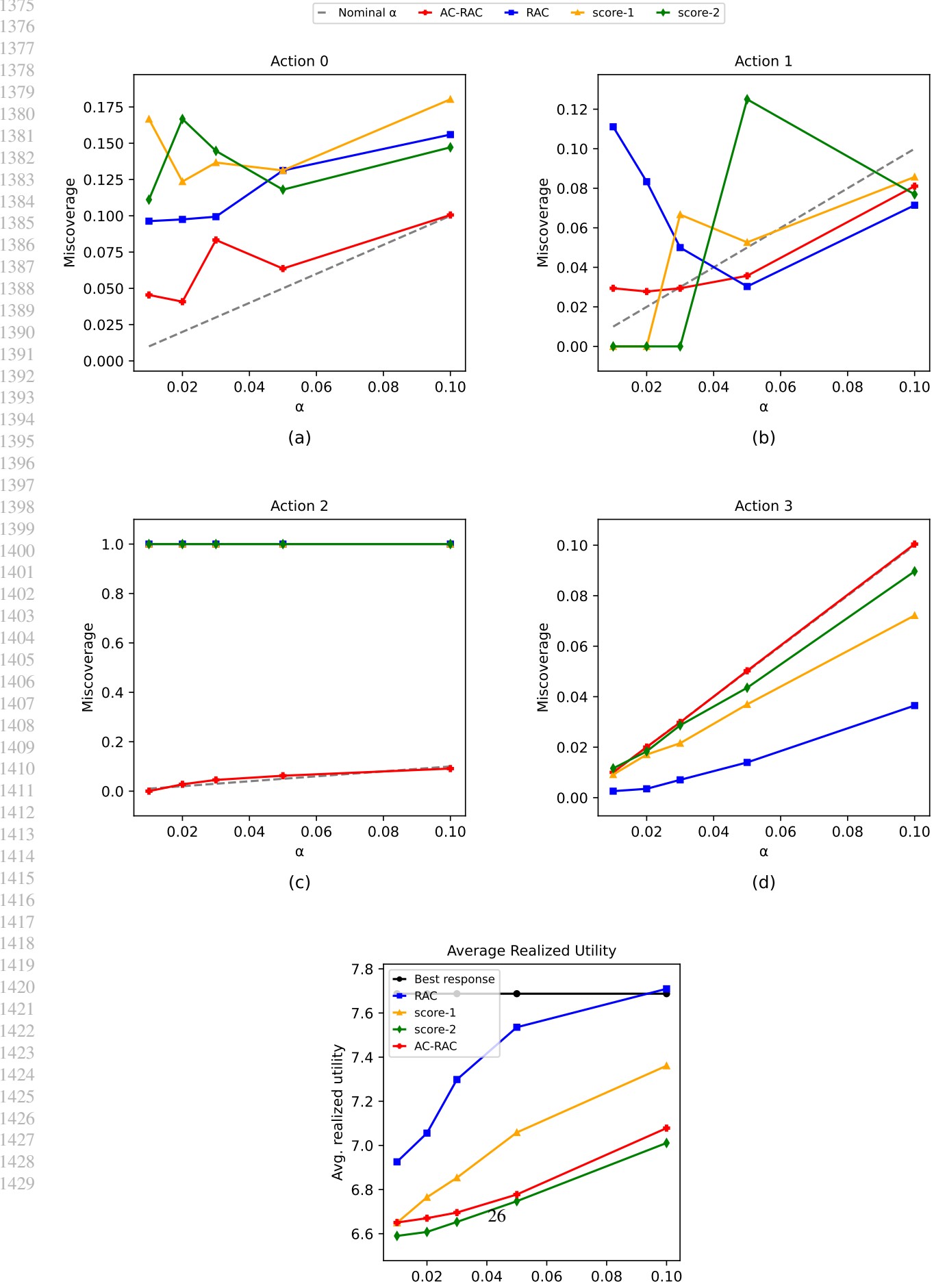

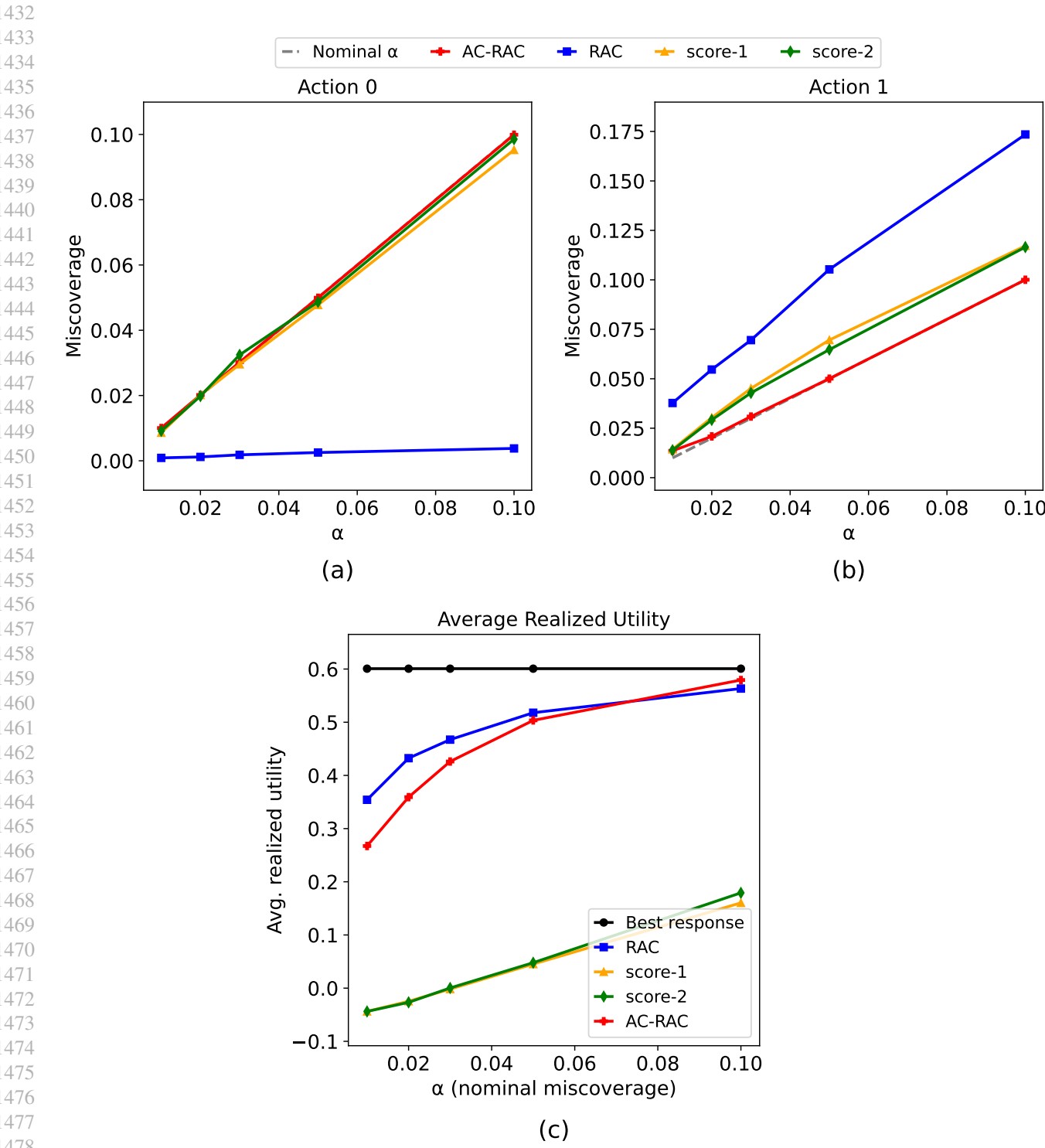

*Figure 3.* (a–b) Miscoverage vs. $\alpha$ for each action under AC-RAC (red), RAC (blue), s1 (orange), s2 (green) (dashed identity). (c) Average utility vs. $\alpha$.