# OpenReview forum: "Conformal Risk-Averse Decision Making with Action Conditional Guarantee"
_ICML.cc/2026/Conference — ICML 2026 regular_

### Official Review · Reviewer_6UYc · 2026-03-10

**Soundness:** 3
**Presentation:** 3
**Significance:** 3
**Originality:** 3
**Overall Recommendation:** 4
**Confidence:** 4

**Summary:**

The authors propose an approach for risk-averse decision-making via action-conditional conformal prediction. Specifically, they propose the action-conditional risk averse (AC-RAC) framework to guarantee label coverage. Specifically, they apply conditional conformal prediction on the action selected by the policy. However, this policy is determined by the conformal prediction. Thus, the authors are required to handle this coupling using the pinball loss formulation (page 6).

**Compliance With Llm Reviewing Policy:**

Affirmed.

**Final Justification:**

The authors have addressed my concerns, and I have already appropriately increased my score to a more positive level.

**Key Questions For Authors:**

(See Weaknesses):
1. How does the prediction set size get affected by applying the action-conditioning?
2. How does the set size change when the action space is larger?

**Limitations:**

Yes

**Strengths And Weaknesses:**

**Strengths**

- The authors provide a theoretically sound formulation to provide action-conditioned guarantees. Furthermore, they formulate a differentiable optimization problem to provide the calibrated thresholds. Lastly, the empirical results validate their conditional coverage guarantees (which the baselines don’t meet).

**Weaknesses**

- As is standard in conformal prediction, the prediction set size is an important metric to quantify its utility, especially as a predictive or decision-making mechanism. In this regard, the authors need to provide the prediction set size and compare it to the size of prediction sets from vanilla conformal prediction (without the action-conditioned guarantees).

- The data sets used have a limited number of actions (no more than 3). The authors should evaluate how their approach fares on datasets with more actions and how this impacts the prediction set size compared to vanilla conformal prediction.

I will reconsider the score if these additional experiments are included.

---

> ### Author Rebuttal · Authors · 2026-03-30
>
> We sincerely thank the reviewer for the thoughtful and constructive feedback. We especially appreciate the two concrete suggestions on (i) reporting prediction-set size and (ii) testing the effect of a larger action space. We agree these are important for assessing the practical value of our method, and we have now run both experiments. Since your overall assessment indicated that additional empirical evidence could meaningfully strengthen the paper, we focused our rebuttal on providing exactly these missing results.
>
> ### W1. As is standard in ...
>
> We fully agree. In the original submission we emphasized coverage/control properties, but for any set-valued predictor, the average set size is a key measure of informativeness. For our method, this question is particularly important because action-conditioning strengthens the guarantee from a marginal statement to an action-conditional one, and the natural concern is whether this stronger control is achieved only by making the output sets much larger.
>
> ### Q1. How does the prediction set size ...
>
> We now report the mean prediction-set size on the COVID task across nominal levels:
>
> | $\alpha$ | AC-RAC | RAC | Score-1 | Score-2 |
> |---:|---:|---:|---:|---:|
> | 0.01 | 3.797 | 3.773 | 2.980 | 2.957 |
> | 0.02 | 3.701 | 3.646 | 2.802 | 2.812 |
> | 0.03 | 3.566 | 3.487 | 2.664 | 2.681 |
> | 0.05 | 3.373 | 3.252 | 2.468 | 2.419 |
> | 0.10 | 2.964 | 2.750 | 2.102 | 2.058 |
>
> The main takeaway is that the cost of action-conditioning is visible but modest relative to RAC. For example, at $\alpha=0.05$, the mean set size increases from $3.252$ to $3.373$, which is only about a 3.7\% increase. Relative to standard score-based conformal baselines, the increase is larger, but this is expected: those methods are calibrated only marginally and may obtain smaller sets partly by avoiding harder actions.
>
> Conceptually, this is the same trade-off that appears when moving from marginal to finer-grained conditional guarantees in conformal prediction. In our setting, AC-RAC pays a limited increase in set size in exchange for ensuring that the downstream decision rule does not disproportionately exclude rare or difficult actions. We believe this is a reasonable utility/safety trade-off for risk-aware decision making.
>
> ### W2. The data sets used ...
>
> We agree that this should be tested empirically rather than addressed only at a qualitative level. Our theory is stated for a finite discrete action set, but an important practical question is whether the method remains usable when $|\mathcal A|$ becomes moderately larger.
>
> ### Q2. How does the set size change ...
>
> To address this directly, we performed a controlled scaling ablation by enlarging the utility matrix to create 6, 8, and 10 non-dominated actions on the same COVID score distribution. We report a compact highlight table below, summarizing prediction-set size, coverage, critical error, and action usage:
>
> | $\|\mathcal A\|$ | Method | Mean prediction set size | Overall coverage | Critical error rate | Actions used |
> |---:|---|---:|---:|---:|---:|
> | 4  | AC-RAC | 3.373 | 0.946 | 0.35% | 4 |
> | 4  | RAC    | 3.252 | 0.953 | 3.99% | 3 |
> | 6  | AC-RAC | 3.042 | 0.968 | 0.21% | 6 |
> | 6  | RAC    | 3.128 | 0.957 | 3.35% | 5 |
> | 8  | AC-RAC | 3.643 | 0.981 | 0.21% | 8 |
> | 8  | RAC    | 3.136 | 0.951 | 4.02% | 6 |
> | 10 | AC-RAC | 3.415 | 0.967 | 0.21% | 9 |
> | 10 | RAC    | 3.187 | 0.950 | 4.70% | 9 |
>
> These results are encouraging in three ways. First, the mean prediction set size stays in a fairly narrow range rather than exploding with $|\mathcal A|$, suggesting that action-conditioning does not incur rapidly growing conservativeness in this regime. Second, AC-RAC continues to keep nearly all available actions active, whereas RAC uses fewer actions when the space is smaller, which is consistent with our goal of avoiding collapse onto only the dominant actions. Third, the core safety message of the paper also persists: AC-RAC maintains a very low critical error rate (0.21--0.35\%) across all tested action-space sizes, while RAC remains substantially worse on this metric and in fact becomes slightly worse as more actions are introduced (3.99\% to 4.70\%). This indicates that the benefit of action-conditional calibration is not limited to the smallest action spaces.
>
> We emphasize that this is a controlled synthetic stress test rather than a claim that large, continuous, or combinatorial action spaces are already solved, since the additional actions come from a hand-crafted enlarged utility matrix rather than a new real-world dataset. We report only a compact summary here due to rebuttal space, and will include the full detailed scaling results in the revised manuscript. Even with this caveat, we believe the experiment directly addresses the reviewer’s tractability concern and shows that the main empirical conclusions of the paper remain intact as the action space grows.

---

> > ### Author Rebuttal · Reviewer_6UYc · 2026-04-01
> >
> > All of my concerns were addressed. Please include these new results in the updated manuscript, as they are meaningful. Furthermore, as noted by the other reviewers, please clearly scope his work as operating on **discrete** action spaces. I'd even recommend briefly discussing how one may get this to work on continuous action spaces
> >
> > I will update my score from a 3 to a 4

---

### Official Review · Reviewer_iNpE · 2026-03-11

**Soundness:** 3
**Presentation:** 3
**Significance:** 2
**Originality:** 2
**Overall Recommendation:** 4
**Confidence:** 4

**Summary:**

This paper extends recent work on risk-averse decision-making using Conformal Prediction (CP) by transitioning from marginal to action-conditional safety guarantees. While previous frameworks ensure that decisions are safe on average across a population , this paper proposes a method (AC-RAC) that guarantees high-utility outcomes conditionally on each specific action chosen by the agent. The authors formulate an Action-Conditional Conformal Prediction Optimization (AC-CPO) problem, show it serves as a feasible surrogate for Action-Conditional Decision Policy Optimization (AC-DPO) , and solve it via a reparameterized dual formulation. The proposed finite-sample algorithm minimizes the pinball loss of action-specific nonconformity scores. Empirical results on medical diagnosis and recommender systems demonstrate that AC-RAC satisfies conditional coverage where marginal methods fail.

**Compliance With Llm Reviewing Policy:**

Affirmed.

**Final Justification:**

Most of my concerns have been addressed so I increased my score.

**Key Questions For Authors:**

1. Can you provide an example to clarify in what specific scenario does the relaxation of AC-CPO become overly conservative?
2. Is it possible to optimize $F_y(\lambda)$ efficiently over a continuous domain?
3. How does AC-RAC behave empirically when an action is selected only 1% or 0.1% of the time in the calibration set?

**Limitations:**

The authors are encouraged to address the identified weaknesses and questions to improve the paper.

**Strengths And Weaknesses:**

**Strengths:**

 1. Moving from marginal to action-conditional safety is a highly critical step for deploying machine learning in high-stakes environments (e.g., healthcare).
 2. Formulating the calibration step as a pinball loss minimization over a continuous dual variable $\lambda$ (Algorithm 1) is a creative way to enforce exact empirical quantiles for conditional sub-populations.
 3. The theoretical finite-sample validity bounds are solid.

**Weaknesses:**

 1. There are many approximation or relaxation issues. First, theorem 2.1 only guarantees a one-directional correspondence (AC-CPO induces a feasible AC-DPO policy). Because the conditioning event is endogenous to the prediction set, AC-CPO is a relaxation. Second, the gradient ddescent on the optimization for $F_{y}(\lambda)$ for each $y$.

 2. The dual parameter vector $\lambda$ is defined as having dimension $|\mathcal{A}|$. Consequently, the framework is strictly limited to small, discrete action spaces. It cannot be applied to continuous control problems (e.g., autonomous driving, robotics) or combinatorial action spaces.

---

> ### Author Rebuttal · Authors · 2026-03-30
>
> We sincerely thank the reviewer for the thoughtful and constructive feedback. Below we address each question directly.
>
> ### W1. There are many ...
>
> We agree the one-directional statement is the key technical subtlety. In AC-CPO, the conditioning event $\{a^{\widehat C}_{RA}(X)=a\}$ is induced by the prediction set itself, whereas in AC-DPO the policy $a(X)$ is free. The main difficulty is that the prediction set now plays two roles at once: it determines which action is chosen and also determines the subpopulation on which coverage must be certified. This feedback makes the action-conditional problem harder than the marginal one and is exactly where the relaxation enters.
>
> We also acknowledge that there may be a more elegant formulation of AC-CPO that is equivalent to AC-DPO, and we will discuss this as an interesting future-work direction.
>
> On optimization, we use subgradient descent because $F_y(\lambda)$ is suitable for first-order optimization but has no simple closed-form solution. By Lemma 4.1, each sample enters through the scalar comparison $\lambda_a \ge \lambda_a^*$, so evaluating a subgradient only requires computing the induced actions and the action-wise empirical coverage terms. This makes each iteration simple and scalable in the number of actions. Appendix C.4 gives the convergence statement. In practice, optimization was stable across experiments.
>
> ### W2/Q2. The dual parameter ... / Is it possible to ...
>
> While the current paper focuses on discrete action spaces, the framework could in principle be extended to continuous domains. We see two natural extensions:
>
> 1. **Discretization:** discretize the action space and apply AC-RAC on the discretized action set. This preserves the current finite-sample guarantee on that discretization.
> 2. **Function approximation:** parameterize $\lambda(a)$ using a class such as linear features, kernels, or a neural network, and optimize the corresponding pinball objective over that class.
>
> The second direction is promising, but the current guarantee would not transfer verbatim to the continuous setting. New arguments would be needed to control the complexity of the function class and the induced action-dependent conditioning events. We will make this limitation and the current scope much more explicit in the revision.
>
> We also conducted additional experiments on scaling up the action set. In a controlled study with 4, 6, 8, and 10 actions, we observe that: prediction-set size stays in a narrow range, AC-RAC continues to use most or all available actions, and its critical error remains very low, while RAC remains worse on this safety metric. We report and discuss the detailed results in our responses to Reviewers 6oy9 and 6UYc, and will include the full scaling results in the revision.
>
> ### Q1. Can you provide an example ...
>
> Consider two actions: a safe default $a_0$ with moderate utility across all labels, and a specialized action $a_1$ with very high utility on a small subset of inputs but poor utility elsewhere. A free AC-DPO policy may choose $a_1$ only when the predictive distribution is concentrated and revert to $a_0$ otherwise. But AC-CPO must certify coverage conditional on the action induced by the conformal set, so points near the boundary between $a_0$ and $a_1$ can force larger sets or more fallback to $a_0$ than an unconstrained policy would choose.
>
> Empirically, this conservativeness appears modest on COVID: at $\alpha=0.05$, AC-RAC attains average utility 6.795 vs. 7.558 for RAC, while its average realized max-min certificate is 6.085 vs. 6.376. Thus the price of action-conditional calibration is modest relative to the safety gain.
>
> We remark that this is also where Theorem 4.3 is informative: the slack term scales like
> $|\mathcal A|/[(n+1)\mathbb P(a^{C_{\mathrm{final}}}_{RA}(X)=a)]$,
> so the most delicate regime is when an action is very rare. This motivated the rare-action stress test below.
>
>
>
> ### Q3. How does AC-RAC behave empirically ...
>
> We added a targeted stress test by downsampling the calibration points associated with a chosen action until its prevalence drops from 5.3% to 1%, 0.5%, and 0.1%.
>
> | Target prevalence in calibration | Method | Target-action coverage | Overall coverage | Mean prediction set size |
> |---:|---|---:|---:|---:|
> | 5.3% (full) | AC-RAC | 0.927 | 0.946 | 3.373 |
> | 5.3% (full) | RAC | 0.871 | 0.953 | 3.252 |
> | 1.0% | AC-RAC | 0.934 | 0.945 | 3.350 |
> | 1.0% | RAC | 0.871 | 0.952 | 3.242 |
> | 0.5% | AC-RAC | 0.931 | 0.945 | 3.341 |
> | 0.5% | RAC | 0.871 | 0.952 | 3.242 |
> | 0.1% | AC-RAC | 0.934 | 0.945 | 3.349 |
> | 0.1% | RAC | 0.871 | 0.952 | 3.242 |
>
> The main takeaway is that AC-RAC does not collapse when the action becomes rare: its target-action coverage remains around 0.93--0.94 and its mean prediction-set size stays essentially unchanged. We will present this carefully as a controlled stress test, since the predictive scores are fixed and only the calibration prevalence is perturbed.

---

> > ### Author Rebuttal · Reviewer_iNpE · 2026-04-03
> >
> > Thanks for the detailed feedback. Most of my concerns have been addressed and I will increase my score. Good luck!

---

### Official Review · Reviewer_6oy9 · 2026-03-12

**Soundness:** 4
**Presentation:** 3
**Significance:** 4
**Originality:** 4
**Overall Recommendation:** 5
**Confidence:** 4

**Summary:**

This work introduces a novel framework, AC-RAC that yields action-conditional guarantees in finite samples. It improves upon existing conformal prediction methods that only promise marginal safety. The effectiveness of the algorithm and improvements over baselines is demonstrated on two real-world experiments.

**Compliance With Llm Reviewing Policy:**

Affirmed.

**Key Questions For Authors:**

(1) For the experiments, it may be useful to include the false discovery rate (FDR) of AC-RAC compared to baselines? The current results illustrate the improvement of AC-RAC over baselines in the miscoverage rate and critical error rate, however, this could come at the cost of being too conservative. Hence, highlighting the FDR could further substantiate these results.

(2) In figure 1, the left plot only contains actions 0, 1, and 3. Is there any explanation behind this - is action 2 an extremely high-risk action? Is it possible that the methods are being overly conservative?

(3) Are there any restrictions on the properties of the action space A (size, discreteness, etc.), the highlighted experiments have rather small action spaces, is this a requirement for tractability or just a coincidence?

**Limitations:**

Yes. The authors discuss potential extensions to other notions of safety.

**Strengths And Weaknesses:**

Soundness: This work is technically sound with well-supported claims with theoretical and empirical results. The authors are careful and honest about evaluating both the strengths and weaknesses of their work.

Presentation: The paper is clearly written and well structured, and the narrative easy to follow. The work positions itself in the context of existing literature and clearly discusses how it differs.

Significance: The paper addresses the issue of conditional coverage guarantees in conformal prediction. It advances the understanding and paves the way for future work in conditional coverage guarantees that can broadly impact the field.

Originality: This work provides new insights of action-conditional conformal prediction that highlights the importance of obtaining conditional coverage guarantees. These contributions are clearly distinguished from closely related literature while building upon existing
frameworks.

---

> ### Author Rebuttal · Authors · 2026-03-30
>
> We sincerely thank the reviewer for the positive assessment and for highlighting the importance of action-conditional guarantees. We also appreciate the concrete suggestions. Below we address each question directly.
>
> ### Q1. For the experiments, it may be ...
>
> Yes. We agree that this is an important diagnostic. We define FDR in the set-prediction sense: for a prediction set $C(X)$ and true label $Y$, the false discovery proportion is $∣C(X)∖\{Y\}∣/∣C(X)∣$, and FDR is its expectation. We computed the false discovery rate (FDR) at $\alpha=0.05$ on the COVID task.
>
> | Method | Mean prediction set size | Overall FDR |
> |---|---:|---:
> | AC-RAC | 3.373 | 0.699 |
> | RAC | 3.252 | 0.682 |
> | Score-1 | 2.468 | 0.585 |
> | Score-2 | 2.419 | 0.550 |
>
> The key point is that AC-RAC is not obtaining stronger guarantees through a dramatic increase in conservativeness. Relative to RAC, its mean set size increases only from 3.252 to 3.373 (about 3.7%), while the overall FDR increases only modestly (0.699 vs. 0.682), consistent with the small change in set size. In other words, the price of action-conditional calibration is visible but modest.
>
> It is also important to interpret FDR jointly with the action frequencies. Some baselines achieve lower FDR partly because they avoid difficult actions rather than calibrating them well. For example, Score-1 selects Antibiotics only 6 times out of 4234 test instances (0.14%) and never selects Quarantine, so its per-action FDR on such actions is unstable or undefined. We will add the FDR table in the supplement and clarify this interpretation in the text.
>
> ### Q2. In figure 1, the left plot ...
>
> The reason action 2 is missing is that the baselines never choose it on this task, so their action-conditional error for that action is undefined. The test-time action frequencies are:
>
> | Method | No Action | Antibiotics | Quarantine | Additional Testing |
> |---|---:|---:|---:|---:|
> | AC-RAC | 4.87% | 1.11% | 2.72% | 91.31% |
> | RAC | 30.94% | 0.68% | 0.00% | 68.38% |
> | Score-1 | 4.72% | 0.14% | 0.00% | 95.13% |
> | Score-2 | 12.45% | 0.52% | 0.00% | 87.03% |
> | Best Response | 44.45% | 1.82% | 0.00% | 53.73% |
>
> Thus, AC-RAC is the only method that uses Quarantine at all (115 of 4234 test points). RAC, Score-1, Score-2, and the unconstrained best-response policy never select that action. Action 2 is omitted simply because the baselines never select it, so their conditional error for that action is undefined. We will make this explicit in the revised caption/text so the absence of action 2 in the original plot is not confusing.
>
> More broadly, this is consistent with the phenomenon our method is designed to address: a marginally calibrated or unconstrained method can look acceptable on average while effectively avoiding rare or difficult actions. AC-RAC keeps each action in the calibration loop rather than allowing the procedure to concentrate only on the dominant action.
>
> ### Q3. Are there any restrictions ...
>
> Our current theory and algorithm assume a finite action space. Concretely, AC-RAC calibrates an action-specific threshold vector $\lambda$, so the present guarantee is stated for discrete $\mathcal A$.
>
> That said, the small action spaces in our experiments are not only a tractability artifact; they also reflect the target applications we study, where the action set is naturally discrete (e.g., diagnostic choices). To probe tractability, we ran a controlled scaling ablation on the COVID task by enlarging the utility matrix from 4 to 6, 8, and 10 non-dominated actions. We report a summary below, highlighting prediction-set size, coverage, critical error, and action usage:
>
> | $\|\mathcal A\|$ | Method | Mean Prediction set size | Overall coverage | Critical error rate | Actions used |
> |---:|---|---:|---:|---:|---:|
> | 4  | AC-RAC | 3.373 | 0.946 | 0.35% | 4 |
> | 4  | RAC    | 3.252 | 0.953 | 3.99% | 3 |
> | 6  | AC-RAC | 3.042 | 0.968 | 0.21% | 6 |
> | 6  | RAC    | 3.128 | 0.957 | 3.35% | 5 |
> | 8  | AC-RAC | 3.643 | 0.981 | 0.21% | 8 |
> | 8  | RAC    | 3.136 | 0.951 | 4.02% | 6 |
> | 10 | AC-RAC | 3.415 | 0.967 | 0.21% | 9 |
> | 10 | RAC    | 3.187 | 0.950 | 4.70% | 9 |
>
> These results suggest that the main empirical picture remains the same as the action space grows in this controlled finite-action regime. First, the mean prediction set size stays in a fairly narrow range rather than blowing up with $|\mathcal A|$, indicating that action-conditioning does not incur rapidly growing conservativeness in this setting. Second, the core safety message also persists: AC-RAC maintains very low critical error rates (0.21--0.35\%) across all tested action-space sizes, while RAC remains substantially worse on this metric (3.35--4.70\%). Third, AC-RAC continues to use most or all available actions (4/4, 6/6, 8/8, 9/10), which is consistent with our goal of avoiding collapse onto only the dominant actions and preserving meaningful action diversity even as the action space expands.

---

> > ### Author Rebuttal · Reviewer_6oy9 · 2026-04-03
> >
> > I believe the authors addressed all my inquiries and their reply indicates they were perceptive of the feedback, I maintain my positive evaluation.

---

### Decision · Program_Chairs · 2026-04-30

**Decision:**

Accept (regular)

**Comment:**

The paper introduces an action-conditional conformal prediction framework (AC-RAC) for risk-averse decision-making, moving beyond standard marginal safety guarantees. All reviewers agreed that the paper tackles a highly critical problem for high-stakes ML deployments and found the proposed methodology to be technically sound, original, and well-presented.

During the discussion phase, the reviewers raised valid practical concerns, primarily regarding the potential inflation of prediction set sizes, the method's behavior when handling rare actions, and its current limitation to small, discrete action spaces. The authors provided a highly effective and thorough rebuttal. They supplied new empirical data demonstrating that the increase in prediction set size and False Discovery Rate (FDR) is modest. Furthermore, they included a controlled stress test showing robustness on rare actions (down to 0.1% prevalence) and a scaling ablation confirming stable performance as the action space grows. The reviewers unanimously agreed that their concerns were fully resolved and maintained or raised their scores in favor of acceptance.

Overall, this is a technically solid paper that advances the community's understanding of reliable decision-making and uncertainty quantification. It clearly meets the acceptance criteria for ICML.

As agreed upon during the review process, the authors must incorporate the following into the final manuscript:
1. Explicitly scope the work to discrete action spaces early in the text and include the promised discussion on potential extensions to continuous domains.
2. Integrate all the new empirical results provided in the rebuttal, including the FDR tables, prediction set size comparisons, rare-action stress tests, and the action-space scaling ablations (either in the main text or the appendix as space permits).